# Riemannian SAM: Sharpness-Aware Minimization on Riemannian Manifolds

**Jihun Yun**
KAIST
arcprime@kaist.ac.kr

**Eunho Yang**
KAIST, AITRICS
eunhoy@kaist.ac.kr

## Abstract

Contemporary advances in the field of deep learning have embarked upon an exploration of the underlying geometric properties of data, thus encouraging the investigation of techniques that consider general manifolds, for example, hyperbolic or orthogonal neural networks. However, the optimization algorithms for training such geometric deep models still remain highly under-explored. In this paper, we introduce Riemannian SAM by generalizing conventional Euclidean SAM to Riemannian manifolds. We successfully formulate the sharpness-aware minimization on Riemannian manifolds, leading to one of a novel instantiation, Lorentz SAM. In addition, SAM variants proposed in previous studies such as Fisher SAM can be derived as special examples under our Riemannian SAM framework. We provide the convergence analysis of Riemannian SAM under a less aggressively decaying ascent learning rate than Euclidean SAM. Our analysis serves as a theoretically sound contribution encompassing a diverse range of manifolds, also providing the guarantees for SAM variants such as Fisher SAM, whose convergence analyses are absent. Lastly, we illustrate the superiority of Riemannian SAM in terms of generalization over previous Riemannian optimization algorithms through experiments on knowledge graph completion and machine translation tasks.

## 1 Introduction

Deep learning incorporating the underlying geometry of data, referred to as geometric deep learning (GDL), has emerged as a significant research area in recent years due to its remarkable capability to effectively capture intrinsic structural properties. As a significant direction in this line, hyperbolic representation learning has been shown to offer several advantages over conventional Euclidean geometry. For example, the hyperbolic space allows for more efficient representations of high-dimensional data by offering a more flexible and natural way to model hierarchical structures, which are commonly encountered in network embeddings [1, 2, 3], computer visions [4, 5], and natural language processing [6, 7]. Adding to the fascinating array of approaches in geometric deep learning, the orthogonal neural networks enforcing the orthogonality in model parameters emerge as a promising research area. In another dimension, the orthogonal neural networks that constrain the model parameter to satisfy the orthogonality (known as Stiefel manifold) are proposed [8, 9, 10] under the motivation that they prevent the vanishing/exploding gradient problem and theoretically enhance the model generalization [11]. Furthermore, more generally, the Riemannian extension of deep learning technique on Euclidean space continues to be proposed in many fields including Riemannian normalizing flows [12, 13], Riemannian diffusion models [14], Poincaré ResNet [15], hyperbolic deep reinforcement learning [16].

Along with the attempts to learn non-Euclidean representations in deep learning, Riemannian optimization has also been greatly studied to train non-Euclidean deep models. As a pioneering example, Riemannian (stochastic) gradient descent (R(S)GD) [17] is an extension of (stochastic) gradient descent to Riemannian manifolds, which updates the gradient computed on (a random subset of) the

37th Conference on Neural Information Processing Systems (NeurIPS 2023).

training data at each iteration and then projects the gradient onto the tangent space of the manifold before taking a descent step. Starting with Riemannian gradient descent, several popular optimization algorithms in Euclidean space, such as conjugate gradient and trust region, have been generalized to Riemannian manifolds [18, 19, 20, 21, 22, 23, 24]. In addition to these previous works, there have been many studies to incorporate the momentum and variance reduction technique into the Riemannian manifolds. In this line of work, many optimization algorithms are proposed including the stochastic variance reduction scheme (R-SVRG) [25], stochastic recursive momentum (R-SRG) [26], and stochastic path-integrated differential estimator (R-SPIDER) [27], extended from their Euclidean counterparts. Going beyond the first-order algorithms, Riemannian (quasi)-Newton methods [28, 29] are a family of second-order methods using a (quasi)-Newton approach on Riemannian manifolds. In the context of deep learning, Riemannian extensions of adaptive gradient methods (ex. AdaGrad/Adam/AMSGrad) are proposed [30, 31, 32], which combines the benefits of adaptive learning rate methods with the efficiency of Riemannian optimization techniques.

However, the exploration of optimization algorithms for training deep learning models on non-Euclidean geometry has been considerably limited, highlighting the need to generalize successful optimizers in Euclidean space to Riemannian manifolds. Under this motivation, we introduce a new class of optimization schemes on Riemannian manifolds. Toward this, as our motivating optimization algorithm, we consider the sharpness-aware minimization (SAM) [33] in Euclidean space, which efficiently improves model generalization by considering the underlying geometry of loss landscapes. With the great success of SAM, several SAM variants including adaptive SAM [34], Fisher SAM [35], Efficient SAM [36], and GSAM [37], are proposed in recent years. In this paper, we propose Riemannian SAM, a sharpness-aware minimization on Riemannian manifolds that can be applied to various manifolds. Our Riemannian SAM considers the sharpness of the loss defined on manifolds, thereby effectively improving model generalization. We believe that our framework could further bring out the potential of the Riemannian deep models and enable more accurate evaluation.

Our contributions are summarized as follows:

- We introduce Riemannian SAM, a sharpness-aware minimization scheme on Riemannian manifolds. Under our framework, we present a novel instance, Lorentz SAM, mainly used in our empirical evaluations. Furthermore, one of the SAM variants, Fisher SAM, which considers the underlying distribution space of neural networks can be derived as a special example under our Riemannian SAM framework.

- We provide the convergence analysis of Riemannian SAM. Our convergence analysis achieves the first-order optimal rate of SGD and we highlight the challenges in our analysis. We allow for a less aggressively decaying ascent learning rate than the condition in the convergence of Euclidean SAM. Also, we provide the convergence guarantee for the SAM variants such as Fisher SAM whose convergence proofs are absent.

- We validate the Riemannian SAM on knowledge graph completion and machine translation tasks for hyperbolic neural networks. The state-of-the-art hyperbolic architecture equipped with our Riemannian SAM improves the performance of the baselines trained with Riemannian Adam, which is a conventional optimizer in Riemannian deep learning.

## 2 Preliminaries

Before introducing the sharpness-aware minimization on Riemannian manifolds, we organize the necessary concepts and notations for Riemannian geometry and Riemannian optimization.

### 2.1 Riemannian Geometry for Optimization

We refer to the definitions in literature [38, 39, 40] where one can find more details.

**Definition 1** (Riemannian manifold). *For each $w \in \mathcal{M}$, let $\mathrm{T}_w\mathcal{M}$ denote the **tangent space** at $w$. An **inner product** on tangent space $\mathrm{T}_w\mathcal{M}$ is a bilinear, symmetric, positive definite function $g_w(\cdot, \cdot) := \langle \cdot, \cdot \rangle_w : \mathrm{T}_w\mathcal{M} \times \mathrm{T}_w\mathcal{M} \to \mathbb{R}$. If a metric $\langle \cdot, \cdot \rangle_w$ smoothly varies with $w \in \mathcal{M}$, we call $\langle \cdot, \cdot \rangle_w$ a **Riemannian metric**. An induced norm on $\mathrm{T}_w\mathcal{M}$ is $\|\zeta\|_w := \sqrt{\langle \zeta, \zeta \rangle_w}$. A **Riemannian manifold** is a pair $(\mathcal{M}, g)$ of the manifold $\mathcal{M}$ and the associated Riemannian metric tensor $g$.*

**Definition 2** (Geodesic). *A **geodesic** is a curve $\gamma(\cdot) : [0,1] \to \mathcal{M}$ that locally minimizes the distance between two points on a manifold with constant speed, which is the generalization of a straight line in Euclidean space.*

**Definition 3** (Exponential maps/Retraction). *An **exponential map** $\exp_w : \mathrm{T}_w\mathcal{M} \to \mathcal{M}$ maps a tangent vector $\zeta_w \in \mathrm{T}_w\mathcal{M}$ onto $\mathcal{M}$ along a geodesic curve such that $\gamma(0) = w$ and $\gamma(1) = z$ with $\dot{\gamma}(0) = \zeta_w$. Specifically, $\gamma(t) := \exp_w(\zeta_w t)$ represents a geodesic. A **retraction** $\mathrm{R}_w(\cdot)$ is a (computationally efficient) generalization of an exponential map satisfying the following properties:*

- *$\mathrm{R}_w(0) = w$ and $D\mathrm{R}_w(0) = \mathrm{Id}_{\mathrm{T}_w\mathcal{M}}$ where $D\mathrm{R}_w$ represents the derivatives of $\mathrm{R}_w$ and $\mathrm{Id}_{\mathrm{T}_w\mathcal{M}}$ denotes an identity map on $\mathrm{T}_w\mathcal{M}$.*

**Definition 4** (Parallel translation/Vector transport). *A **parallel translation** $P_z^w(\cdot) : \mathrm{T}_z\mathcal{M} \to \mathrm{T}_w\mathcal{M}$ transport a tangent vector in $\mathrm{T}_z\mathcal{M}$ to $\mathrm{T}_w\mathcal{M}$ in parallel while preserving norm and direction (i.e., along a geodesic). A **vector transport** $\mathcal{T}(\gamma)_z^w(\cdot) : \mathrm{T}_z\mathcal{M} \to \mathrm{T}_w\mathcal{M}$ **with respect to retraction map** $\mathrm{R}$ maps a vector $\zeta_z \in \mathrm{T}_z\mathcal{M}$ to $\zeta_w \in \mathrm{T}_w\mathcal{M}$ along a retraction curve $\gamma(t) = \mathrm{R}_w(\xi_w t)$ for some $\xi_w \in \mathrm{T}_w\mathcal{M}$, which is computationally efficient approximation of a parallel translation. In this work, we only consider **isometric** vector transport, i.e., $\langle \mathcal{T}_z^w\zeta_z, \mathcal{T}_z^w\eta_z \rangle_w = \langle \zeta_w, \eta_w \rangle_w$ for all $\zeta_w, \eta_w \in \mathrm{T}_w\mathcal{M}$.*

We introduce important examples of Riemannian manifolds in deep learning. The initial two instances represent dominant manifolds within the realm of hyperbolic deep learning. Hyperbolic space is a natural geometry for capturing underlying tree-like, graph-shaped, or hierarchical structures, which are properties existing in many real datasets. Owing to this characteristic, there have been many approaches to hyperbolic deep learning encompassing network embeddings [1, 2, 3, 41], computer vision [4, 5], and natural language processing [42, 6]. For manifolds in hyperbolic space, we mainly follow the definitions in [1, 2].

**Poincaré Ball.** The Poincaré ball $\mathbb{P}^n = (\mathbb{B}^n, g_p)$ is a Riemannian manifold with

$$\mathbb{B}^n = \{x \in \mathbb{R}^n : \|x\|_2 < 1\}, \quad g_p(x) = \left(\frac{2}{1 - \|x\|_2^2}\right)^2 g_e(x).$$

where $g_e$ represents an Euclidean metric tensor. The associated distance on Poincaré ball is given by

$$d_p(x, y) = \mathrm{arcosh}\left(1 + 2\frac{\|x - y\|_2^2}{(1 - \|x\|_2^2)(1 - \|y\|_2^2)}\right).$$

We remark on some properties of the Poincaré ball. The Poincaré ball is a conformal model, meaning that angles between curves are preserved under conformal transformations. This property enables the Poincaré ball to accurately represent the local geometry of complex spaces. Additionally, the Poincaré ball offers an intuitive visualization of hyperbolic spaces, particularly in two or three dimensions. Despite these advantageous properties, the Poincaré ball also presents challenges in computing mathematical concepts such as geodesics and distances.

**Lorentz Model.** The Lorentz model $\mathbb{L}^n = (\mathbb{H}^n, g_\ell)$ is a semi-Riemannian manifold, but it is still possible to employ Riemannian optimization. The Lorentz model consist of

$$\mathbb{H}^n = \{x \in \mathbb{R}^{n+1} : \langle x, x \rangle_\mathbb{L} = -1, x_0 > 0\}, \quad g_\ell(x) = \begin{bmatrix} -1 & \cdots & 0 & 0 \\ 0 & 1 & \cdots & 0 \\ \vdots & \vdots & \ddots & \vdots \\ 0 & 0 & \cdots & 1 \end{bmatrix}. \tag{1}$$

where $\langle x, y \rangle_\mathbb{L} = -x_0 y_0 + \sum_{j=1}^n x_j y_j$ is known as the *Lorentzian scalar product*. The associated distance function is given by

$$d_\ell(x, y) = \mathrm{arcosh}\left(-\langle x, y \rangle_\mathbb{L}\right)$$

Note that an $n$-dimensional Lorentz model requires one more redundant dimension in Euclidean space. Importantly, Poincaré ball and the Lorentz model are equivalent under the diffeomorphism $\varphi : \mathbb{H}^n \to \mathbb{P}^n$ (with the corresponding inverse mapping $\varphi^{-1} : \mathbb{P}^n \to \mathbb{H}^n$) defined as

$$\varphi(x_0, x_1, \cdots, x_n) = \frac{(x_1, x_2, \cdots, x_n)}{p_0 + 1}, \tag{2}$$

$$\varphi^{-1}(x_1, x_2, \cdots, x_n) = \frac{(1 + x_1^2 + \cdots + x_n^2, 2x_1, \cdots, 2x_n)}{1 - (x_1^2 + \cdots + x_n^2)}. \tag{3}$$

We also remark on some characteristics of the Lorentz model. The main advantage is that it allows for more stable Riemannian optimization under relatively simple formulas for mathematical quantities, such as geodesics and distances. However, the visualization is difficult due to less intuitive projection onto a lower-dimensional space. As noted in each manifold, both two manifolds on hyperbolic space are equivalent, but they have different purposes. For this reason, some studies [2] use Lorentz manifolds for their model design and training, then visualize the results on Poincaré ball using the diffeomorphism $\varphi$ in (2) and (3).

**Stiefel manifold.** The orthogonality, i.e., $W^{\mathsf{T}}W = I$ for model parameter $W$, plays a role to circumvent the vanishing/exploding gradient problem [8, 10, 43] and delivers theoretically enhanced generalization error bounds [11]. The Stiefel manifold $\mathrm{St}(n, p) = (\mathbb{V}^n, g_W)$ for $n \geq p$ is prevalent for orthogonal neural networks, which is also a Riemannian manifold defined by

$$\mathbb{V}^n = \{X \in \mathbb{R}^{n \times p} : X^{\mathsf{T}}X = I\}, \quad g_W(Z_1, Z_2) = \mathrm{Tr}(Z_1^{\mathsf{T}}Z_2).$$

for tangent vectors $Z_1, Z_2 \in \mathrm{T}_W\mathrm{St}(n, p)$. The tangent space of $\mathrm{St}(n, p)$ at $W$ is defined by $\mathrm{T}_W\mathrm{St}(n, p) = \{Z : Z^{\mathsf{T}}W + W^{\mathsf{T}}Z = 0\}$.

## 2.2 Riemannian Optimization

We are interested in the following optimization problem over the Riemannian manifold $(\mathcal{M}, g)$

$$\min_{w \in \mathcal{M}} \mathcal{L}(w).$$

where $\mathcal{L} : \mathcal{M} \to \mathbb{R}$ is a smooth function defined on manifold $\mathcal{M}$. Following the work [17], Riemannian stochastic gradient descent (RSGD) updates the model parameter $w \in \mathcal{M}$ as

$$w_{t+1} = \exp_{w_t}\big(-\alpha_t \mathrm{grad}\mathcal{L}(w_t)\big). \tag{4}$$

where $\mathrm{grad}\mathcal{L}(w_t) \in \mathrm{T}_{w_t}\mathcal{M}$ is a Riemannian gradient at $w_t$ and $\alpha_t$ is the learning rate. In some practical cases, the exponential map is computationally inefficient. Hence, it may be replaced with (more computationally efficient) suitable retraction map $\mathrm{R}_{w_t}(\cdot)$, yielding the update rule $w_{t+1} = \mathrm{R}_{w_t}\big(-\alpha_t \mathrm{grad}\mathcal{L}(w_t)\big)$. Generally, the Riemannian gradient $\mathrm{grad}\mathcal{L}(w_t)$ in (4) is computed with the Riemannian metric tensor $g$ as

$$\mathrm{grad}\mathcal{L}(w_t) = g^{-1}(w_t)\nabla\mathcal{L}(w_t). \tag{5}$$

where $\nabla\mathcal{L}(w_t)$ denotes the Euclidean gradient. This is also known as natural gradient descent [44]. The quantity on the right-hand side in (5) may not be on $\mathrm{T}_{w_t}\mathcal{M}$. In this case, we should project the gradient onto the tangent space since the exponential map is not defined.

## 3 Sharpness-Aware Minimization on Riemannian Manifolds

In empirical risk minimization (ERM) including deep learning tasks, we generally minimize the finite-sum objective (or equivalently expected objective) for training dataset $\mathcal{D} = \{(x_i, y_i)\}_{i=1}^n$ as

$$\min_{w \in \mathcal{M}} \mathcal{L}(w) := \frac{1}{n} \sum_{i=1}^n \mathcal{L}(w; x_i). \tag{6}$$

where the smooth loss function $\mathcal{L}(\cdot) : \mathcal{M} \to \mathbb{R}$ is defined on a Riemannian manifold $\mathcal{M}$. We formulate the sharpness-aware minimization in terms of loss function values on the manifold $\mathcal{M}$ as

$$\min_{w \in \mathcal{M}} \max_{\|\delta\|_w^2 \leq \rho^2} \underbrace{\left\{\mathcal{L}(\mathrm{R}_w(\delta)) - \mathcal{L}(w)\right\}}_{\text{Sharpness in terms of loss function}}. \tag{7}$$

In contrast to Euclidean space, the Riemannian manifold is not a vector space in general. Hence, the familiar concepts defined in Euclidean space may not be well-defined. Therefore, we restrict the perturbation $\delta$ in the tangent space $\mathrm{T}_w\mathcal{M}$. To solve the inner subproblem, we resolve the inner optimization problem in a different manner as

$$\max_{\delta \in \mathrm{T}_w\mathcal{M}} \mathcal{L}\big(\mathrm{R}_w(\delta)\big) - \mathcal{L}(w) \quad \text{such that} \quad \|\delta\|_w^2 \leq \rho^2. \tag{8}$$

for a fixed point $w \in \mathcal{M}$. For ease of computations, we approximate the perturbed loss function $\mathcal{L}(\mathrm{R}_w(\delta))$ via Taylor's expansion as

$$\mathcal{L}(\mathrm{R}_w(\delta)) \approx \mathcal{L}(w) + \langle \mathrm{grad}\mathcal{L}(w), \delta \rangle_w. \tag{9}$$

Then, our inner maximization problem comes in hand:

$$\max_{\delta \in \mathrm{T}_w\mathcal{M}} \langle \mathrm{grad}\mathcal{L}(w), \delta \rangle_w \quad \text{such that} \quad \|\delta\|_w^2 \leq \rho^2. \tag{10}$$

This problem could be easily solved since it finds the steepest direction $\delta$ on Riemannian manifold $\mathcal{M}$, whose solution is known to be just Riemannian gradient. Therefore, we have

$$\delta^* = \rho \frac{\mathrm{grad}\mathcal{L}(w)}{\|\mathrm{grad}\mathcal{L}(w)\|_w}. \tag{11}$$

Under the optimal perturbation $\delta^*$ in (11), we further approximate the gradient of the sharpness-aware minimization in (7) (for outer minimization problem) as

$$\mathrm{grad}\mathcal{L}(\mathrm{R}_w(\delta^*)) \approx \mathrm{grad}\mathcal{L}(w)|_{w=\mathrm{R}_w(\delta^*)}. \tag{12}$$

since the left-hand side of (12) requires a higher-order Riemannian gradient, which is not computationally feasible in practice. The remaining is that the approximated Riemannian SAM gradient $\mathrm{grad}\mathcal{L}(w)|_{w=\mathrm{R}_w(\delta^*)}$ in (12) is not on the tangent space at $w$, $\mathrm{T}_w\mathcal{M}$. To perform an actual parameter update on $w$, we should transport the Riemannian SAM gradient to the tangent space $\mathrm{T}_w\mathcal{M}$ via vector transport $\mathcal{T}_{\mathrm{R}_w(\delta^*)}^w$ with respect to the retraction $\mathrm{R}_w$. We summarize the overall optimization procedure in Algorithm 1. Note that, in order to consider the most practical case, we assume that the same minibatch is used for computing SAM perturbation with the ascent step and the actual parameter updates with the descent step (see lines 5, 6, and 8). In fact, one can use different batches for lines $5 \sim 7$ and lines $8 \sim 10$ respectively or full-batch gradient for both ascent and descent steps.

**Remarks on Algorithm 1.** In fact, it might be most natural to choose a perturbation region at the current point as in the conventional Euclidean SAM, $\delta \in B_\rho(w_t) := \{x \in \mathcal{M} : d_{\mathcal{M}}(w_t, x) \leq \rho\}$ where $d_{\mathcal{M}}$ represents the distance on the manifold. However, adopting the constraint in this manner may pose challenges in utilizing the standard assumptions for analyzing non-convex Riemannian optimization, such as geodesic or retraction smoothness (see condition (C-4) in Section 4), which makes it difficult to guarantee convergence. Moreover, the computation of $d_{\mathcal{M}}$ is often computationally inefficient in practice. Another possible extension is to apply the vector transport operation from line 8 of Algorithm 1 to line 9. The following outlines the modified procedure: (i) $g_t^{adv} = \mathcal{A}(\mathrm{grad}\mathcal{L}(w; \mathcal{S})|_{w=w_t^{adv}})$ and (ii) $\Delta_t = \mathcal{T}_{w_t^{adv}}^{w_t} g_t^{adv}$. For base optimizer $\mathcal{A}$, any optimization algorithm commonly used in Riemannian optimization can be adopted (e.g., Riemannian SGD). In the meanwhile, when the vector transport is applied after constructing $g_t^{adv}$ via the momentum-based optimizer $\mathcal{A}$, the momentum construction takes place on the tangent space $\mathrm{T}_{w_t^{adv}}\mathcal{M}$ at the perturbed point $w_t^{adv}$, while the parameter update occurs on the different tangent space at the point. As a consequence, this might introduce another challenges in understanding and analyzing the overall optimization process. In this perspective, various alternative extensions can also be possible, but among them, we have carefully designed a *theoretically valid, computationally practical, and non-trivially extended Sharpness-Aware Minimization on general manifolds for Riemannian optimization*. Then, we have successfully demonstrated both convergence analysis (Section 4) and empirical studies (Section 5) to corroborate our Riemannian SAM.

**Existing example of Riemannian SAM framework: Fisher SAM.** Following our Riemannian SAM update rule in Algorithm 1, we can show that Fisher SAM is a special instance of Riemannian SAM. We can view the set of neural networks as a neuromanifold [45, 46] equipped with the KL divergence metric between two points. Hence, let $w \in \mathcal{M}$ be the point on a neuromanifold (or statistical manifold) $\mathcal{M}$ which is realized by Euclidean network parameter $\theta \in \mathbb{R}^d$. On distribution space, the corresponding metric tensor $g$ is known to be Fisher information matrix [44, 45, 46]. According to Algorithm 1, the perturbation at line 6 and 7 could be computed as

$$\mathrm{grad}\mathcal{L}(w) = F(\theta)^{-1}\nabla\mathcal{L}(\theta)$$

$$\delta^* = \rho \frac{\mathrm{grad}\mathcal{L}(w)}{\|\mathrm{grad}\mathcal{L}(w)\|_w} = \rho \frac{F(\theta)^{-1}\nabla\mathcal{L}(\theta)}{\sqrt{\mathrm{grad}\mathcal{L}(w)^\mathsf{T}F(\theta)\mathrm{grad}\mathcal{L}(w)}} = \rho \frac{F(\theta)^{-1}\nabla\mathcal{L}(\theta)}{\sqrt{\nabla\mathcal{L}(\theta)^\mathsf{T}F(\theta)^{-1}\nabla\mathcal{L}(\theta)}}.$$

---

**Algorithm 1** Riemannian SAM: Sharpness-Aware Minimization on Riemannian Manifolds

---

1: **Input:** Descent learning rate $\alpha_t$, ascent learning rate $\rho_t$, and a base Riemannian optimizer $\mathcal{A}$ (such as Riemannian SGD, Riemannian Adam).
2: **Initialize:** $w_0 \in \mathcal{M}$
3: **for** $t = 0, 1, \ldots, T-1$ **do**
4:    Draw a minibatch sample $\mathcal{S} = \{x_1, \cdots, x_{|\mathcal{S}|}\}$.
5:    $g_t \leftarrow \frac{1}{|\mathcal{S}|} \sum\limits_{i=1}^{|\mathcal{S}|} \text{grad}\mathcal{L}(w_t; x_i).$             ▷ Stochastic Riemannian gradient
6:    $\delta_t \leftarrow \frac{g_t}{\|g_t\|_{w_t}}.$                            ▷ Perturbation (11)
7:    $w_t^{adv} \leftarrow \text{R}_{w_t}(\rho_t \delta_t).$                       ▷ SAM ascent step
8:    $g_t^{adv} \leftarrow \mathcal{T}_{w_t^{adv}}^{w_t} \text{grad}\mathcal{L}(w; \mathcal{S})|_{w=w_t^{adv}}.$      ▷ Gradient at $w_t^{adv}$ and transport to $\text{T}_{w_t}\mathcal{M}$
9:    $\Delta_t^{adv} \leftarrow \mathcal{A}(g_t^{adv}).$               ▷ An update vector via a base optimizer $\mathcal{A}$
10:    $w_{t+1} \leftarrow \text{R}_{w_t}(-\alpha_t \Delta_t^{adv}).$                  ▷ Final descent step
11: **end for**
12: **Output:** $w_T$

---

which is entirely identical to the perturbation of Fisher SAM [35].

**Novel example: Lorentz SAM on hyperbolic geometry.** We derive the novel instance of Riemannian SAM called Lorentz SAM over the Lorentz model introduced in 2.1. First, we derive the Riemannian gradient on the Lorentz model $\mathbb{L}^n = (\mathbb{H}^n, g_\ell)$. As in Section 2.2, the Riemannian gradient could be computed as

$$h = g_\ell^{-1} \nabla \mathcal{L}(w). \tag{13}$$

Since $g_\ell$ in (1) is a diagonal matrix, it is easy to compute the vector $h$ with Euclidean gradient $\nabla \mathcal{L}(w)$. However, the vector $h$ is not on the tangent space at $w$, $\text{T}_w \mathbb{L}^n$, thus we should have to project the vector $h$ onto the tangent space $\text{T}_w \mathbb{L}^n$. The projection is easily computed in a closed-form as

$$\text{proj}_w(v) = v + \langle w, v \rangle_{\mathbb{L}} w. \tag{14}$$

Hence, the Riemannian gradient on Lorentz model is computed by

$$\text{grad}\mathcal{L}(w) = \text{proj}_w\big(g_\ell^{-1} \nabla \mathcal{L}(w)\big). \tag{15}$$

As a next step, we should normalize the Riemannian gradient as in line 6 in Algorithm 1, and this is easy to compute as $\|\text{grad}\mathcal{L}(w)\|_w = \sqrt{\langle \text{grad}\mathcal{L}(w), \text{grad}\mathcal{L}(w) \rangle_{\mathbb{L}}}$ via Lorentzian scalar product $\langle \cdot, \cdot \rangle_{\mathbb{L}}$ defined in Section 2.1.

We utilize the Lorentz SAM derived above for our primary empirical studies conducted on hyperbolic space. In a similar way, one can derive the Riemannian SAM on Poincaré ball or Stiefel manifold, which we defer to Appendix.

### 3.1 Riemannian SAM Illustration: Toy 3D Illustration

We illustrate the 3-dimensional toy example on the sphere manifold. Let us define the 3D sphere manifold $\mathbb{S}^2$ with the tangent space at $w$, $\text{T}_w \mathcal{M}$ as

$$\mathbb{S}^2 := \{w \in \mathbb{R}^3 : \|w\|_2 = 1\},$$
$$\text{T}_w \mathbb{S}^2 := \{v \in \mathbb{R}^3 : w^\mathsf{T} v = 0\}$$

We consider the regression problem with the neural-net-like objective function on the randomly generated synthetic dataset. Toy 3D optimization problem with objective function $f(w) = \frac{1}{2n}\|y - \text{ReLU}(Xw)\|_2^2$ where $X \in \mathbb{R}^{500 \times 3}$ and $y \in \mathbb{R}^{500}$ are drawn from $\mathcal{N}(0, 1^2)$ and $\mathcal{U}(0, 1)$ respectively with the model parameter $w = (x, y, z) \in \mathbb{R}^3$ under $\|w\|_2 = 1$. **(a)** Comparison of converged points for each method. We plot the contour plots with the spherical coordinates under the relation $(x, y, z) \leftrightarrow (r, \theta, \varphi) = (1, \theta, \varphi)$.

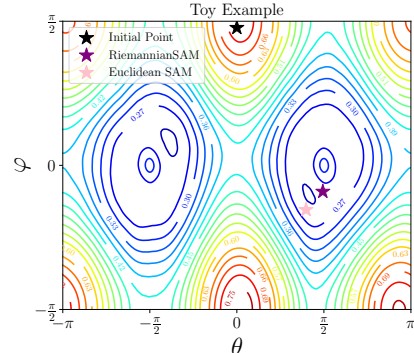

Figure 1: Toy 3D illustration

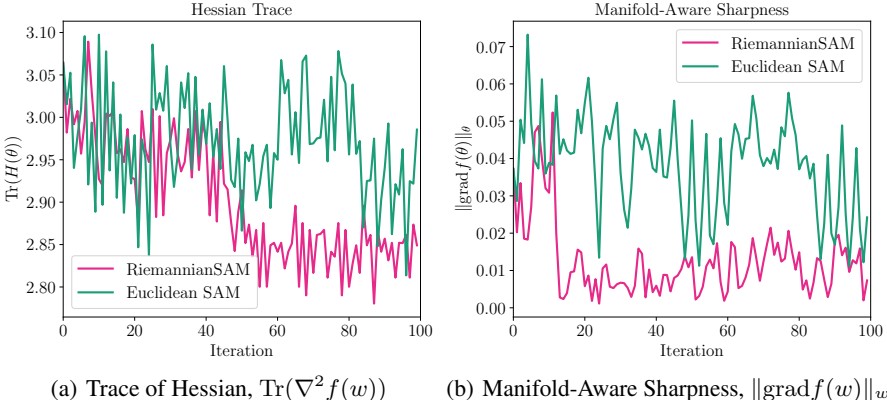

(a) Trace of Hessian, $\mathrm{Tr}(\nabla^2 f(w))$    (b) Manifold-Aware Sharpness, $\|\mathrm{grad} f(w)\|_w$

Figure 2: Comparison of two sharpness measures.

The Figure 1 corresponds to Cartesian coordinates $(x, y, z)$ to spherical coordinates $(r, \theta, \varphi) = (1, \theta, \varphi)$, rendering contour plots. In Figure 1, we showcases the converged points on the objective function under the optimization using Riemannian SAM (in purple color) and the conventional Euclidean SAM (in pink color). Within a maximum iteration budget 100, the purple point (Riemannian SAM) attains a loss value of 0.3800 while the pink point (conventional Euclidean SAM) converges with a slightly higher loss value of 0.3808.

Furthermore, in terms of sharpness measures, we consider the following basic two quantities: (i) the trace of the Hessian (sharpness in the context of Euclidean space), and (ii) manifold-aware sharpness, characterized by the Riemannian gradient norm $\|\mathrm{grad}\mathcal{L}(w)\|_w$. Notably, the manifold-aware sharpness aligns with information-geometric sharpness [47] when dealing with statistical manifolds, where the Riemannian metric is defined by the Fisher information. For the aforementioned problem, we compare two metrics and Figure 2 depicts the results. In both metrics, Riemannian SAM achieves smaller sharpness values than Euclidean SAM, implying convergence toward flatter regions. In other words, since the Euclidean SAM might fail to properly consider the underlying structure of the manifold even for toy examples, this phenomenon is expected to be exacerbated in extremely high-dimensional problems such as deep learning.

## 4 Convergence Analysis

In this section, we present the convergence guarantees for the RiemSAM framework. Our goal is to find a first-order $\epsilon$-approximate solution: the output $\widetilde{w}$ such that $\mathbb{E}\big[\|\mathrm{grad}\mathcal{L}(\widetilde{w})\|_{\widetilde{w}}^2\big] \leq \epsilon^2$, which is a generalized convergence criterion of Euclidean $\epsilon$-stationary point. To guarantee the convergence for $\epsilon$-approximate solution, we require the following mild assumptions.

**(C-1)** (Upper-Hessian bounded) The objective function $\mathcal{L}$ is said to be *upper-Hessian bounded* in $\mathcal{U} \subset \mathcal{M}$ with respect to retraction $\mathrm{R}$ if there exists some positive constant $C$ such that $\frac{d^2\mathcal{L}(\mathrm{R}_w(t\eta))}{dt^2} \leq C$, for all $w \in \mathcal{U}$ and $\eta \in \mathrm{T}_w\mathcal{M}$ with $\|\eta\|_w = 1$, and for all $t$ such that $\mathrm{R}_w(\tau\eta)$ for all $\tau \in [0, t]$.

**(C-2)** (Lower-bounded) The objective function $\mathcal{L}(\cdot)$ is differentiable and has bounded suboptimality.
$$\mathcal{L}(w^*) > -\infty.$$
for the optimal point $w^* \in \mathcal{M}$.

**(C-3)** (Unbiasedness and bounded variance) The stochastic Riemannian gradient is unbiased and has a bounded variance:
$$\mathbb{E}_{(x,y)\in\mathcal{D}}[\mathrm{grad}\mathcal{L}(w; x)] = \mathrm{grad}\mathcal{L}(w),$$
$$\mathbb{E}_{(x,y)\in\mathcal{D}}[\|\mathrm{grad}\mathcal{L}(w; x) - \mathrm{grad}\mathcal{L}(w)\|_w^2] \leq \sigma^2.$$
where $\mathrm{grad}\mathcal{L}(w)$ is a true Riemannian gradient evaluated on a full batch of training dataset $\mathcal{D}$.

**(C-4)** (Retraction smoothness) We assume that there exists a constant $L_S > 0$ such that
$$\mathcal{L}(z) \leq \mathcal{L}(w) + \langle \mathrm{grad}\mathcal{L}(w), \eta \rangle_w + \frac{1}{2}L_S\|\eta\|_w^2.$$

for all $w, z \in \mathcal{M}$ and $\gamma(t) := \mathrm{R}_w(t\eta)$ represents a retraction curve on $\mathcal{M}$ for $\eta \in \mathrm{T}_w\mathcal{M}$ with the starting point $\gamma(0) = w$ and the terminal point $\gamma(1) = z$.

**(C-5)** (Individual Retraction Lipschitzness) We assume that there exists $L_{\mathrm{R}} > 0$ such that

$$\|\mathcal{T}(\gamma)_z^w \mathrm{grad}\mathcal{L}(z;x) - \mathrm{grad}\mathcal{L}(w;x)\|_w \leq L_{\mathrm{R}}\|\eta\|_w.$$

for all $w, z \in \mathcal{M}$. As in condition (C-4), $\gamma(t)$ denotes a retraction curve and $\mathcal{T}(\gamma)_z^w$ is a vector transport associated with this retraction curve.

The function class with condition (C-1) corresponds to the continuous function family with Lipschitz continuous gradients in the Euclidean space [26, 32, 48]. The assumptions (C-2) $\sim$ (C-4) are standard in convergence analysis of Riemannian optimization algorithms, under which Riemannian SGD is known to be first-order optimal [49]. Note that, unlike in Euclidean space, the constant $L_S$ in (C-4) and $L_{\mathrm{R}}$ in (C-5) may be different. According to [26, 50], the condition (C-5) can be derived under the standard assumption on retraction Lipschitzness with parallel translation and one additional assumption on the bound between the parallel translation and the vector transport, but we assume the retraction Lipschitzness with the vector transport for simplicity. Lastly, in condition (C-5), the retraction Lipschitzness is assumed individually with respect to each sample in order to control the alignment of SAM gradient and the original gradient step as in [51].

Now, we are ready to present our main theorem.

**Theorem 1** (Convergence of Riemannian SAM). *Let $\widetilde{w}$ denote an iterate uniformly chosen at random from $\{w_1, w_2, \cdots, w_T\}$. Further, we let $\widetilde{L} = \max\{L_S, L_{\mathrm{R}}\}$ where the constants $L_S$ and $L_{\mathrm{R}}$ are defined in condition (C-4) and (C-5) respectively. Under the conditions (C-1) $\sim$ (C-5) with descent learning rate $\alpha_t = \frac{1}{\sqrt{T}\widetilde{L}}$ and ascent learning rate $\rho_t = \frac{1}{T^{1/6}\widetilde{L}}$, we have the following complexity for the constant batch size $b$:*

$$\mathbb{E}\big[\|\mathrm{grad}\mathcal{L}(\widetilde{w})\|_{\widetilde{w}}^2\big] \leq \frac{Q_1\Delta}{\sqrt{T}} + \frac{Q_2\sigma^2}{b\sqrt{T}} + \frac{Q_3\sigma^2}{bT^{5/6}} = \mathcal{O}(1/\sqrt{T}). \tag{16}$$

*where $\Delta = \mathcal{L}(w_0) - \mathcal{L}(w^*)$ and the constants $\{Q_i\}_{i=1}^3$ are irrelevant to the total iteration $T$ or the manifold dimension $d$.*

We make some remarks on our convergence results and relationship to conventional Euclidean SAM.

**Theoretical implications.** Our key observation of Theorem 1 lies in the *alignment between the Riemannian gradient* $\mathrm{grad}\mathcal{L}(w_t)$ *(line 5 in Algorithm 1) and the Riemannian SAM gradient* $\mathcal{T}_{w_t^{adv}}^{w_t} \mathrm{grad}\mathcal{L}(w_t^{adv})$ *(line 9 in Algorithm 1)* for the perturbed point $w_t^{adv}$, $w_t^{adv}$. The previous study [51] on Euclidean SAM says that Euclidean SAM gradient should be well-aligned with the true gradient step for convergence. Unlike the theoretical claim in [51], we stress that for convergence guarantee those gradients should be *well-aligned within the preconditioned space (by inverse Riemannian metric) regardless of alignment in Euclidean space*. To verify this insight, we directly measure the angles between two vectors with a 2D toy example, illustrating how they align in practice. Toward this, we consider two angles: (i) $\angle(\nabla f(w_t^{adv}), \nabla f(w_t))$ (Euclidean Align-

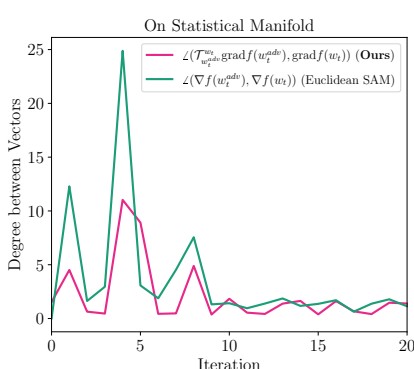

Figure 3: Toy 2D illustration

ment) and (ii) $\angle(\mathcal{T}_{w_t^{adv}}^{w_t} \mathrm{grad}f(w_t^{adv}), \mathrm{grad}f(w_t))$ (Riemannian Alignment, Ours). In this example, we consider the logistic regression where 200 data samples are generated with 100 of them sampled from $\mathcal{N}(-1, 1^2)$ and the remaining sampled from $\mathcal{N}(1, 1^2)$. The labels are assigned such that if a sample was drawn from a Gaussian distribution with a mean of $-1$, the label was set to $y = 0$, and otherwise, we set $y = 1$. We minimize the cross-entropy loss with our Riemannian SAM with the Fisher information matrix as the Riemannian metric. The Figure 3 depicts the comparison of angles. The loss decreases up to 10-th iteration, after which it remains around the converged point. As evident from the illustration, while the angles between the Euclidean space SAM gradient and the gradient deviate by up to around 25 degrees, the angles between the preconditioned SAM gradient and the preconditioned gradient, influenced by the Fisher information, align more closely with deviations

Table 1: Link prediction results (%) in the filterated setting for WN18RR and FB15k-237 datasets. For hyperbolic architectures, $\beta \in \{200, 400, 500\}$ and we report the best result. The results of baselines are taken from [52] except for HYBONET whose results are reproduced by ourselves. The best results among hyperbolic architectures with the same dimensions are in boldface. Our Riemannian SAM (denoted by rSAM) shows the superior performance compared to Riemannian Adam (denoted by rAdam).

| | | WN18RR | | | | FB15k-237 | | | |
|---|---|---|---|---|---|---|---|---|---|
| Manifold | Model | #Dims | MRR | H@10 | H@3 | H@1 | #Dims | MRR | H@10 | H@3 | H@1 |
| | MURP | 32 | 46.5 | 54.4 | 48.4 | 42.0 | 32 | 32.3 | 50.1 | 35.3 | 23.5 |
| | ROTH | 32 | 47.2 | 55.3 | 49.0 | 42.8 | 32 | 31.4 | 49.7 | 34.6 | 22.3 |
| | ATTH | 32 | 46.6 | 55.1 | 48.4 | 41.9 | 32 | 32.4 | 50.1 | 35.4 | 23.6 |
| | HYBONET w/ rAdam | 32 | $48.8^{\pm 0.2}$ | $55.4^{\pm 0.1}$ | $50.3^{\pm 0.2}$ | $45.5^{\pm 0.3}$ | 32 | $33.5^{\pm 0.2}$ | $51.5^{\pm 0.2}$ | $36.5^{\pm 0.3}$ | $24.2^{\pm 0.3}$ |
| Hyperbolic | HYBONET w/ rSAM | 32 | $\mathbf{49.3}^{\pm 0.2}$ | $\mathbf{56.0}^{\pm 0.2}$ | $\mathbf{50.7}^{\pm 0.1}$ | $\mathbf{46.2}^{\pm 0.3}$ | 32 | $\mathbf{34.3}^{\pm 0.2}$ | $\mathbf{52.0}^{\pm 0.1}$ | $\mathbf{37.3}^{\pm 0.3}$ | $\mathbf{25.1}^{\pm 0.4}$ |
| | MURP | $\beta$ | 48.1 | 56.6 | 49.5 | 44.0 | $\beta$ | 33.5 | 51.8 | 36.7 | 24.3 |
| | ROTH | $\beta$ | 49.6 | 58.6 | 51.4 | 44.9 | $\beta$ | 34.4 | 53.5 | 38.0 | 24.6 |
| | ATTH | $\beta$ | 48.6 | 57.3 | 49.9 | 44.3 | $\beta$ | 34.8 | 54.0 | 38.4 | 25.2 |
| | HYBONET w/ rAdam | $\beta$ | $51.2^{\pm 0.2}$ | $57.1^{\pm 0.2}$ | $52.5^{\pm 0.2}$ | $48.3^{\pm 0.2}$ | $\beta$ | 35.2 | 52.9 | 38.7 | 26.3 |
| | HYBONET w/ rSAM | $\beta$ | $\mathbf{51.6}^{\pm 0.2}$ | $\mathbf{58.7}^{\pm 0.3}$ | $\mathbf{53.3}^{\pm 0.2}$ | $\mathbf{48.6}^{\pm 0.1}$ | $\beta$ | $\mathbf{36.0}^{\pm 0.2}$ | $\mathbf{54.3}^{\pm 0.4}$ | $\mathbf{39.6}^{\pm 0.2}$ | $\mathbf{26.6}^{\pm 0.1}$ |

only up to a maximum of 10 degrees. In high-dimensional loss landscapes, we expect that the angles would become significantly larger, corroborating our theoretical insight.

**On upper bound.** Distinct from the convergence of Euclidean SAM [51], our upper bound (16) has the additional term involving $Q_3$, but we still achieve the optimal complexity of SGD, $\mathcal{O}(1/\epsilon^4)$ for $\epsilon$-approximate solution. Note that the presence of term involving the constant $Q_3$ in our bound comes from the fact that (i) smoothness condition (C-4) and Lipschitzness condition (C-5) are not equivalent on manifolds and (ii) we should handle the vector-transported gradients (see line 8 in Algorithm 1) at each iteration, which are the main challenges in our proof. Our results can also provide the guarantees for SAM variants such as Fisher SAM [35], whose convergence guarantees are missing.

## 5 Experiments

We conduct two sets of experiments; (i) knowledge graph completion, and (ii) machine translation. The first experiment aims to evaluate our Riemannian SAM on shallow networks and the second task is for optimizing large-scale deep neural networks. For all our experiments, we consider the Lorentz manifold introduced in Section 2.1 and employ the recent hyperbolic architecture, HYBONET [52]. The HYBONET is a *fully hyperbolic* neural network, whose each layer is constructed on the Lorentz manifold including a linear, attention, residual, and positional encoding layer. We implement our Riemannian SAM upon Geoopt framework [53] written in PyTorch library [54]. Regarding hyperparameters, we basically adhere to the same experiment settings in [52] and the details are provided in each section and Appendix.

### 5.1 Knowledge Graph Completion

A knowledge graph completion aims to predict missing relationships within a knowledge graph, which represents structured information as a collection of entities, their attributes, and the relationships between them. More precisely, knowledge in a graph is of the form of triplets $(h, r, t)$ where $h$, $r$, and $t$ denote the head entity, the relationship or predicate, and the tail entity respectively. In the knowledge graph completion task, given a partially populated knowledge graph, the goal is to predict the missing entity or relationship in a triplet: solving $(h, r, ?)$ and $(?, r, t)$.

In our experiments, we use two popular benchmark datasets; WN18RR [41] and Fb15k-237 [55]. We employ the same data preprocessing in [3] and two standard metrics for evaluations: (i) Mean Reciprocal Rank (MRR), the average of the inverse of the true entity ranking, and (ii) Precision at $K$ (H@K), the proportion of test instances where the correct answer appears in the top-$K$ ranked predictions. For

Table 2: The BLEU scores on the test set of IWSLT '14 and WMT '14 under low-dimensional setting following the hyperbolic study [56] with the word vector dimension $d = 64$. The results on baselines are taken from [52] except for HYBONET whose results are reproduced by ourselves.

| Manifold | Model | IWSLT '14 | WMT '14 |
|---|---|---|---|
| **Euclidean** | CONVSEQ2SEQ [57] | 23.6 | 14.9 |
| | TRANSFORMER [58] | 23.0 | 17.0 |
| **Hyperbolic** | HYPERNN++ [56] | 22.0 | 17.0 |
| | HATT [42] | 23.7 | 18.8 |
| | HYBONET (with Riemannian Adam) | 25.5 | 19.3 |
| | HYBONET (with **Riemannian SAM, Ours**) | **26.0** | **20.1** |

marginal hyperparameter tuning, we tune the ascent learning rate $\rho_t \in \{10^{-4}, 10^{-3}, 10^{-2}, 10^{-1}\}$ for Riemannian SAM and the other hyperparameters are the same as HYBONET [52] for fair comparisons.

Table 1 illustrates the results on WN18RR and Fb15k-237 datasets. As in previous work [52], we test our Riemannian SAM on two different regimes: (i) small embedding dimension 32 and (ii) large dimension $\beta \in \{200, 400, 500\}$ for both datasets. Regarding the large dimension, we report the best results among the dimension candidates. In Table 1, Riemannian SAM on the Lorentz model achieves the best performance with great margins for all comparison metrics. In the same way, Riemannian SAM shows the state-of-the-art performance for all metrics considered under both regimes.

## 5.2 Machine Translation

In this experiment, we evaluate our Riemannian SAM on Lorentz Transformer built with Lorentz components introduced in [52] for IWSLT '14 and WMT '14 benchmark datasets in machine translation. We use the BLEU score as an evaluation metric on the IWSLT '14 test set and the newstest2013 test set of WMT '14 respectively. According to [56], we train hyperbolic models with Riemannian SAM in a low-dimensional setting where the dimension of the word vector is $d = 64$. As in the knowledge graph completion task, we choose the ascent learning rate in $\rho_t \in \{10^{-5}, 10^{-4}, \cdots, 10^{-2}\}$ for marginal hyperparameter tuning.

Table 2 demonstrates the results. HYBONET baseline trained with Riemannian Adam already outperforms the Euclidean Transformer in both IWSLT '14 and WMT '14 datasets. Upon this baseline, we only substitute our Riemannian SAM for Riemannian Adam with other hyperparameters unchanged. As seen in Table 2, Riemannian SAM significantly outperforms the Riemannian Adam baseline for both datasets. Note that both HYPERNN++ and HATT are partially hyperbolic networks, so we could not evaluate our Riemannian SAM on these models since it is difficult for a fair evaluation.

**Wall-clock time.** As in Euclidean SAM, Riemannian SAM requires additional forward and backward propagation in a single iteration loop (see Algorithm 1). Thus, we report the wall-clock time comparison for each experiment. For knowledge graph completion (Section 5.1) and machine translation (Section 5.2), Riemannian SAM takes roughly 1.6 and 1.8 times longer than Riemannian Adam for one epoch, respectively. To alleviate the computational overhead, one can employ the stochastic weight perturbation (SWP) and sharpness-sensitive data selection (SDS) suggested in [36], which do not depend on the manifold structure. Another practical consideration is to use a subset of minibatch in computing perturbation (see line 6 in Algorithm 1) for large-scale models. We leave the study on reducing computational cost as future work.

## 6 Conclusion

In this study, we proposed a sharpness-aware minimization on Riemannian manifolds, called Riemannian SAM. Under our framework, we presented novel examples of Riemannian SAM including a Lorentz SAM. We analyzed the convergence of the Riemannian SAM for general manifolds with a less aggressively decaying ascent learning rate condition. Moreover, we showed that Riemannian SAM can provide the convergence guarantee for SAM variants whose convergence proofs are missing such as Fisher SAM. We also illustrated that Riemannian SAM empirically outperforms ERM-based Riemannian optimization algorithms for popular deep learning tasks with hyperbolic neural networks. As future work, we plan to study the technique to reduce the computations and analyze the generalization error bounds of Riemannian SAM theoretically.

## Acknowledgement

This work was supported by the National Research Foundation of Korea (NRF) grants (No.2018R1A5A1059921, RS-2023-00209060), Institute of Information & Communications Technology Planning & Evaluation (IITP) grants (No.2019-0-00075, Artificial Intelligence Graduate School Program(KAIST)) funded by the Korea government (MSIT).

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

# Supplementary Materials

## A Proofs of Theorem 1

We basically follow the arguments in the convergence of Euclidean SAM [51], but the details are totally different.

**Lemma 1** (Properties of Retraction Smoothness). *Let $w^* = \mathrm{R}_w\big(\rho\mathrm{grad}\mathcal{L}(w)\big)$ and the curve $\gamma(t) = \mathrm{R}_w(t\eta)$ with the endpoints $\gamma(0) = w$ and $\gamma(1) = w^*$. Then, we have*

$$\langle \mathcal{T}(\gamma)^w_{w^*}\mathrm{grad}\mathcal{L}(w^*), \mathrm{grad}\mathcal{L}(w)\rangle \geq (1 - \rho L_\mathrm{R})\|\mathrm{grad}\mathcal{L}(w)\|^2_w$$

*Proof.* By the condition (C-5), we have

$$\|\mathcal{T}(\gamma)^w_{w^*}\mathrm{grad}\mathcal{L}(w^*) - \mathrm{grad}\mathcal{L}(w)\|_w \leq L_\mathrm{R}\|\eta\|_w$$

By the Cauchy-Schwarz inequality, we have

$$|\langle \mathcal{T}(\gamma)^w_{w^*}\mathrm{grad}\mathcal{L}(w^*) - \mathrm{grad}\mathcal{L}(w), \eta\rangle_w| \leq \|\mathcal{T}(\gamma)^w_{w^*}\mathrm{grad}\mathcal{L}(w^*) - \mathrm{grad}\mathcal{L}(w)\|_w\|\eta\|_w$$
$$\leq L_\mathrm{R}\|\eta\|^2_w$$
$$\leq \rho^2 L_\mathrm{R}\|\mathrm{grad}\mathcal{L}(w)\|^2_w$$

Therefore, we obtain

$$\langle \mathcal{T}(\gamma)^w_{w^*}\mathrm{grad}\mathcal{L}(w^*) - \mathrm{grad}\mathcal{L}(w), \rho\mathrm{grad}\mathcal{L}(w)\rangle_w \geq -\rho^2 L_\mathrm{R}\|\mathrm{grad}\mathcal{L}(w)\|^2_w$$

Removing the constant $\rho$, the above inequality becomes

$$\langle \mathcal{T}(\gamma)^w_{w^*}\mathrm{grad}\mathcal{L}(w^*) - \mathrm{grad}\mathcal{L}(w), \mathrm{grad}\mathcal{L}(w)\rangle_w \geq -\rho L_\mathrm{R}\|\mathrm{grad}\mathcal{L}(w)\|^2_w$$

Lastly, we arrive at the final result as

$$\langle \mathcal{T}(\gamma)^w_{w^*}\mathrm{grad}\mathcal{L}(w^*), \mathrm{grad}\mathcal{L}(w)\rangle_w = \langle \mathcal{T}(\gamma)^w_{w^*}\mathrm{grad}\mathcal{L}(w^*) - \mathrm{grad}\mathcal{L}(w), \mathrm{grad}\mathcal{L}(w)\rangle_w + \|\mathrm{grad}\mathcal{L}(w)\|^2_w$$
$$\geq (1 - \rho L_\mathrm{R})\|\mathrm{grad}\mathcal{L}(w)\|^2_w$$

$\square$

In the next lemma, we will show the alignment of the true Riemannian gradient and the true Riemannian SAM gradient.

**Lemma 2** (Alignment of the true Riemannian gradient and the true Riemannian SAM gradient). *Let us denote the stochastic Riemannian gradient at time $t$ by $\mathrm{grad}\mathcal{L}_t(w) = \frac{1}{b}\sum_{i\in J_t}\mathrm{grad}\mathcal{L}(w; x_i) \in \mathrm{T}_w\mathcal{M}$ and $w^{adv} = \mathrm{R}_w\big(\rho\mathrm{grad}\mathcal{L}_t(w)\big)$. Further, let $\gamma(t) = \mathrm{R}_w(t\eta)$ be a retraction curve with $\gamma(0) = w$ and $\gamma(1) = w^{adv}$. Then, we have the following inequality*

$$\mathbb{E}\Big[\Big\langle \mathcal{T}(\gamma)^w_{w^{adv}}\mathrm{grad}\mathcal{L}_t(w^{adv}), \mathrm{grad}\mathcal{L}(w)\Big\rangle_w\Big] \geq \Big(\frac{1}{2} - \rho L_\mathrm{R} - 3\rho^2 L_\mathrm{R}^2\Big)\Big\|\mathrm{grad}\mathcal{L}(w)\Big\|^2_w - \frac{2\rho^2 L_\mathrm{R}^2\sigma^2}{b}$$

*Proof.* Let $w^* = \mathrm{R}_w\big(\rho\mathrm{grad}\mathcal{L}(w)\big)$ evaluated on the loss function. We first add and subtract $\langle \mathcal{T}(\zeta)^w_{w^*}\mathrm{grad}\mathcal{L}_t(w^*), \mathrm{grad}\mathcal{L}(w)\rangle_w$ where $\zeta(t) = \mathrm{R}_w(t\xi)$ is a retraction curve where $\zeta(0) = w$ and $\zeta(1) = w^*$.

$$\langle \mathcal{T}(\gamma)^w_{w^{adv}}\mathrm{grad}\mathcal{L}_t(w^{adv}), \mathrm{grad}\mathcal{L}(w)\rangle_w = \underbrace{\langle \mathcal{T}(\gamma)^w_{w^{adv}}\mathrm{grad}\mathcal{L}_t(w^{adv}) - \mathcal{T}(\zeta)^w_{w^*}\mathrm{grad}\mathcal{L}_t(w^*), \mathrm{grad}\mathcal{L}(w)\rangle_w}_{T_1}$$
$$+ \underbrace{\langle \mathcal{T}(\zeta)^w_{w^*}\mathrm{grad}\mathcal{L}_t(w^*), \mathrm{grad}\mathcal{L}(w)\rangle_w}_{T_2}$$

We will bound two terms, $T_1$ and $T_2$, separately. Regarding the term $T_1$, we derive

$$-T_1 = -\langle \mathcal{T}(\gamma)^w_{w^{adv}}\mathrm{grad}\mathcal{L}_t(w^{adv}) - \mathcal{T}(\zeta)^w_{w^*}\mathrm{grad}\mathcal{L}_t(w^*), \mathrm{grad}\mathcal{L}(w)\rangle_w$$
$$\leq \frac{1}{2}\Big\|\mathcal{T}(\gamma)^w_{w^{adv}}\mathrm{grad}\mathcal{L}_t(w^{adv}) - \mathcal{T}(\zeta)^w_{w^*}\mathrm{grad}\mathcal{L}_t(w^*)\Big\|^2_w + \frac{1}{2}\Big\|\mathrm{grad}\mathcal{L}(w)\Big\|^2_w$$
$$\leq \Big\|\mathcal{T}(\gamma)^w_{w^{adv}}\mathrm{grad}\mathcal{L}_t(w^{adv}) - \mathrm{grad}\mathcal{L}_t(w)\Big\|^2_w + \Big\|\mathcal{T}(\zeta)^w_{w^*}\mathrm{grad}\mathcal{L}_t(w^*) - \mathrm{grad}\mathcal{L}_t(w)\Big\|^2_w + \frac{1}{2}\Big\|\mathrm{grad}\mathcal{L}(w)\Big\|^2_w$$
$$\leq L_\mathrm{R}^2\|\rho\mathrm{grad}\mathcal{L}_t(w)\|^2_w + L_\mathrm{R}^2\|\rho\mathrm{grad}\mathcal{L}(w)\|^2_w + \frac{1}{2}\|\mathrm{grad}\mathcal{L}(w)\|^2_w$$
$$\leq \rho^2 L_\mathrm{R}^2\Big(2\|\mathrm{grad}\mathcal{L}_t(w) - \mathrm{grad}\mathcal{L}(w)\|^2_w + 2\|\mathrm{grad}\mathcal{L}(w)\|^2_w\Big) + \Big(\frac{1}{2} + \rho^2 L_\mathrm{R}^2\Big)\|\mathrm{grad}\mathcal{L}(w)\|^2_w$$
$$\leq \frac{2\rho^2 L_\mathrm{R}^2\sigma^2}{b} + \Big(\frac{1}{2} + 3\rho^2 L_\mathrm{R}^2\Big)\|\mathrm{grad}\mathcal{L}(w)\|^2_w$$

From the above inequality, we could finally bound the term $T_1$ as

$$T_1 \geq -\frac{2\rho^2 L_{\mathrm{R}}^2 \sigma^2}{b} - \left(\frac{1}{2} + 3\rho^2 L_{\mathrm{R}}^2\right)\|\mathrm{grad}\mathcal{L}(w)\|_w^2$$

Regarding the term $T_2$, we just use the lemma as

$$T_2 = \langle \mathcal{T}(\zeta)_{w^*}^w \mathrm{grad}\mathcal{L}_t(w^*), \mathrm{grad}\mathcal{L}(w)\rangle_w \geq (1 - \rho L_{\mathrm{R}})\|\mathrm{grad}\mathcal{L}(w)\|_w^2$$

Hence, we arrive at

$$\mathbb{E}\left[\left\langle \mathcal{T}(\gamma)_{w^{adv}}^w \mathrm{grad}\mathcal{L}_t(w^{adv}), \mathrm{grad}\mathcal{L}(w)\right\rangle_w\right] \geq \left(\frac{1}{2} - \rho L_{\mathrm{R}} - 3\rho^2 L_{\mathrm{R}}^2\right)\left\|\mathrm{grad}\mathcal{L}(w)\right\|_w^2 - \frac{2\rho^2 L_{\mathrm{R}}^2 \sigma^2}{b}$$

$\square$

According to Algorithm 1, we follow the notation as

$$\mathrm{grad}\mathcal{L}_t(w) = \frac{1}{b}\sum_{i \in I_t} \mathrm{grad}\ell_i(w)$$

$$w_t^{adv} = \mathrm{R}_{w_t}\left(\rho\mathrm{grad}\mathcal{L}_t(w_t)\right)$$

We assume the stochastic $m$-SAM where the same batch is used for both inner and outer updates.

**Lemma 3** (Descent inequality). *Under the assumptions in Theorem 1, we have*

$$\mathbb{E}\left[\mathcal{L}(w_{t+1})\right] \leq \mathbb{E}\left[\mathcal{L}(w_t)\right] - \frac{3\alpha}{8}\mathbb{E}\left[\|\mathrm{grad}\mathcal{L}(w_t)\|_{w_t}^2\right] + \frac{\alpha^2 L_S \sigma^2}{b} + \frac{2\alpha^2 L_S^3 \rho^2 \sigma^2}{b} + \frac{2\alpha\rho^3 L_{\mathrm{R}}^2 \sigma^2}{b}$$

*Proof.* Using the condition (C-4), we have

$$\mathcal{L}(w_{t+1}) = \mathcal{L}\left(\mathrm{R}_{w_t}\left(-\alpha\mathcal{T}(\gamma)_{w_t^{adv}}^{w_t}\mathrm{grad}\mathcal{L}_t(w_t^{adv})\right)\right)$$

$$\leq \mathcal{L}(w_t) - \alpha\left\langle \mathrm{grad}\mathcal{L}(w_t), \mathcal{T}(\gamma)_{w_t^{adv}}^{w_t}\mathrm{grad}\mathcal{L}_t(w_t^{adv})\right\rangle_{w_t} + \frac{\alpha^2 L_S}{2}\left\|\mathcal{T}(\gamma)_{w_t^{adv}}^{w_t}\mathrm{grad}\mathcal{L}_t(w_t^{adv})\right\|_{w_t}^2$$

For the last term in RHS, we can bound as

$$\left\|\mathcal{T}(\gamma)_{w_t^{adv}}^{w_t}\mathrm{grad}\mathcal{L}_t(w_t^{adv})\right\|_{w_t}^2 = -\|\mathrm{grad}\mathcal{L}(w_t)\|_{w_t}^2 + \left\|\mathcal{T}(\gamma)_{w_t^{adv}}^{w_t}\mathrm{grad}\mathcal{L}_t(w_t^{adv}) - \mathrm{grad}\mathcal{L}(w_t)\right\|_{w_t}^2$$

$$+ 2\left\langle \mathcal{T}(\gamma)_{w_t^{adv}}^{w_t}\mathrm{grad}\mathcal{L}_t(w_t^{adv}), \mathrm{grad}\mathcal{L}(w_t)\right\rangle_{w_t}$$

Again, we have

$$\mathcal{L}(w_{t+1}) \leq \mathcal{L}(w_t) - \alpha\left\langle \mathrm{grad}\mathcal{L}(w_t), \mathcal{T}(\gamma)_{w_t^{adv}}^{w_t}\mathrm{grad}\mathcal{L}_t(w_t^{adv})\right\rangle_{w_t} + \frac{\alpha^2 L_S}{2}\left\|\mathcal{T}(\gamma)_{w_t^{adv}}^{w_t}\mathrm{grad}\mathcal{L}_t(w_t^{adv})\right\|_{w_t}^2$$

$$= \mathcal{L}(w_t) - \frac{\alpha^2 L_S}{2}\|\mathrm{grad}\mathcal{L}(w_t)\|_{w_t}^2 + \frac{\alpha^2 L_S}{2}\left\|\mathcal{T}(\gamma)_{w_t^{adv}}^{w_t}\mathrm{grad}\mathcal{L}_t(w_t^{adv}) - \mathrm{grad}\mathcal{L}(w_t)\right\|_{w_t}^2$$

$$- \alpha(1 - \alpha L_S)\left\langle \mathcal{T}(\gamma)_{w_t^{adv}}^{w_t}\mathrm{grad}\mathcal{L}_t(w_t^{adv}), \mathrm{grad}\mathcal{L}(w_t)\right\rangle_{w_t}$$

$$\leq \mathcal{L}(w_t) - \frac{\alpha^2 L_S}{2}\|\mathrm{grad}\mathcal{L}(w_t)\|_{w_t}^2 + \alpha^2 L_S\left\|\mathcal{T}(\gamma)_{w_t^{adv}}^{w_t}\mathrm{grad}\mathcal{L}_t(w_t^{adv}) - \mathrm{grad}\mathcal{L}_t(w_t)\right\|_{w_t}^2$$

$$+ \alpha^2 L_S\|\mathrm{grad}\mathcal{L}_t(w_t) - \mathrm{grad}\mathcal{L}(w_t)\|_{w_t}^2 - \alpha(1 - \alpha L_S)\left\langle \mathcal{T}(\gamma)_{w_t^{adv}}^{w_t}\mathrm{grad}\mathcal{L}_t(w_t^{adv}), \mathrm{grad}\mathcal{L}(w_t)\right\rangle_{w_t}$$

$$\leq \mathcal{L}(w_t) - \frac{\alpha^2 L_S}{2}\|\mathrm{grad}\mathcal{L}(w_t)\|_{w_t}^2 + \alpha^2 L_S^3 \rho^2\|\mathrm{grad}\mathcal{L}_t(w_t)\|_{w_t}^2 + \alpha^2 L_S\|\mathrm{grad}\mathcal{L}_t(w_t) - \mathrm{grad}\mathcal{L}(w_t)\|_{w_t}^2$$

$$- \alpha(1 - \alpha L_S)\left\langle \mathcal{T}(\gamma)_{w_t^{adv}}^{w_t}\mathrm{grad}\mathcal{L}_t(w_t^{adv}), \mathrm{grad}\mathcal{L}(w_t)\right\rangle_{w_t}$$

$$\leq \mathcal{L}(w_t) - \frac{\alpha^2 L_S}{2}\|\mathrm{grad}\mathcal{L}(w_t)\|_{w_t}^2 + 2\alpha^2 L_S^3 \rho^2\|\mathrm{grad}\mathcal{L}(w_t)\|_{w_t}^2 + 2\alpha^2 L_S^3 \rho^2\|\mathrm{grad}\mathcal{L}_t(w_t) - \mathrm{grad}\mathcal{L}(w_t)\|_{w_t}^2$$

$$+ \alpha^2 L_S\|\mathrm{grad}\mathcal{L}_t(w_t) - \mathrm{grad}\mathcal{L}(w_t)\|_{w_t}^2 - \alpha(1 - \alpha L_S)\left\langle \mathcal{T}(\gamma)_{w_t^{adv}}^{w_t}\mathrm{grad}\mathcal{L}_t(w_t^{adv}), \mathrm{grad}\mathcal{L}(w_t)\right\rangle_{w_t}$$

$$= \mathcal{L}(w_t) - \frac{\alpha^2 L_S(1 - 4L_S^2\rho^2)}{2}\|\mathrm{grad}\mathcal{L}(w_t)\|_{w_t}^2 + \alpha^2 L_S(1 + 2L_S^2\rho^2)\|\mathrm{grad}\mathcal{L}_t(w_t) - \mathrm{grad}\mathcal{L}(w_t)\|_{w_t}^2$$

$$- \alpha(1 - \alpha L_S)\left\langle \mathcal{T}(\gamma)_{w_t^{adv}}^{w_t}\mathrm{grad}\mathcal{L}_t(w_t^{adv}), \mathrm{grad}\mathcal{L}(w_t)\right\rangle_{w_t}$$

Taking the expectation on both sides, we have

$$
\mathbb{E}\big[\mathcal{L}(w_{t+1})\big] \leq \mathbb{E}\big[\mathcal{L}(w_t)\big] - \frac{\alpha^2 L_S(1 - 4L_S^2\rho^2)}{2}\mathbb{E}\big[\|\mathrm{grad}\mathcal{L}(w_t)\|_{w_t}^2\big] + \frac{\alpha^2 L_S(1 + 2L_S^2\rho^2)\sigma^2}{b}
$$

$$
- \alpha(1 - \alpha L_S)\mathbb{E}\left[\Big\langle \mathcal{T}(\gamma)_{w_t^{adv}}^{w_t}\mathrm{grad}\mathcal{L}_t(w_t^{adv}), \mathrm{grad}\mathcal{L}(w_t)\Big\rangle_{w_t}\right]
$$

$$
\mathbb{E}\big[\mathcal{L}(w_t)\big] - \frac{\alpha^2 L_S(1 - 4L_S^2\rho^2)}{2}\mathbb{E}\big[\|\mathrm{grad}\mathcal{L}(w_t)\|_{w_t}^2\big] + \frac{\alpha^2 L_S(1 + 2L_S^2\rho^2)\sigma^2}{b}
$$

$$
- \alpha(1 - \alpha L_S)\left[\Big(\frac{1}{2} - \rho L_{\mathrm{R}} - 3\rho^2 L_{\mathrm{R}}^2\Big)\mathbb{E}\big[\|\mathrm{grad}\mathcal{L}(w_t)\|_{w_t}^2\big] - \frac{2\rho^2 L_{\mathrm{R}}^2\sigma^2}{b}\right]
$$

For sufficiently large number of total iteration $T$, the condition $\rho \leq \frac{1}{4\widetilde{L}}$ is easily satisfied where $\widetilde{L} = \max\{L_{\mathrm{R}}, L_S\}$ (defined in Theorem 1). Hence, we obtain

$$
\frac{3\alpha}{8}\mathbb{E}\big[\|\mathrm{grad}\mathcal{L}(w_t)\|_{w_t}^2\big] \leq \mathbb{E}\big[\mathcal{L}(w_t)\big] - \mathbb{E}\big[\mathcal{L}(w_{t+1})\big] + \frac{\alpha^2 L_S(1 + 2L_S^2\rho^2)\sigma^2}{b} + \frac{2\alpha(1 - \alpha L_S)\rho^3 L_{\mathrm{R}}^2\sigma^2}{b}
$$

$$
\leq \mathbb{E}\big[\mathcal{L}(w_t)\big] - \mathbb{E}\big[\mathcal{L}(w_{t+1})\big] + \frac{\alpha^2 L_S\sigma^2}{b} + \frac{2\alpha^2 L_S^3\rho^2\sigma^2}{b} + \frac{2\alpha\rho^3 L_{\mathrm{R}}^2\sigma^2}{b}
$$

By telescoping the above inequality from $t = 0 \sim T - 1$, we arrive at

$$
\mathbb{E}\big[\|\mathrm{grad}\mathcal{L}(\widetilde{w})\|_{\widetilde{w}}^2\big] = \frac{1}{T}\sum_{t=0}^{T-1}\mathbb{E}\big[\|\mathrm{grad}\mathcal{L}(w_t)\|_{w_t}^2\big] \leq \frac{8\Delta}{3\alpha T} + \frac{8\alpha^2 L_S\sigma^2}{3b} + \frac{16\alpha^2 L_S^3\rho^2\sigma^2}{3b} + \frac{16\alpha\rho^3 L_{\mathrm{R}}^2\sigma^2}{3b}
$$

Under the step size condition $\alpha_t = \frac{1}{\sqrt{T}\widetilde{L}}$ and $\rho_t = \frac{1}{T^{1/6}\widetilde{L}}$, we finally get

$$
\mathbb{E}\big[\|\mathrm{grad}\mathcal{L}(\widetilde{w})\|_{\widetilde{w}}^2\big] \leq \frac{Q_1\widetilde{L}\Delta}{\sqrt{T}} + \frac{Q_2\sigma^2}{b\sqrt{T}} + \frac{Q_3\sigma^2}{bT^{5/6}}
$$

for appropriate constants $\{Q_i\}_{i=1}^3$. $\qquad \square$

# B Hyperparameter Details

We use the almost same hyperparameters in the study [52] and implement our experiments in Section 5 upon its official implementation. For completeness, we summarize the hyperparameter configurations in Table 3 and Table 4.

Table 3: Hyperparameter configurations for knowledge graph completion.

|  | WN18RR | | FB15k-237 | |
| Dimension | 32 | $\beta$ | 32 | $\beta$ |
| --- | --- | --- | --- | --- |
| Batch Size | 1000 | 1000 | 500 | 500 |
| Negative Samples | 50 | 50 | 50 | 50 |
| Margin | 8.0 | 8.0 | 8.0 | 8.0 |
| Epochs | 1000 | 1000 | 500 | 500 |
| Max Norm | 1.5 | 2.5 | 1.5 | 1.5 |
| Max Scaler | 3.5 | 2.5 | 2.5 | 2.5 |
| Learning Rate | 0.005 | 0.003 | 0.003 | 0.003 |
| Gradient Norm | 0.5 | 0.5 | 0.5 | 0.5 |

Table 4: Hyperparameter configurations for machine translation.

| Hyperparameter | IWSLT'14 | WMT'14 |
| --- | --- | --- |
| GPU Numbers | 4 | 4 |
| Embedding Dimension $d$ | 64 | 64 |
| Feed-forward Dimension | 256 | 256 |
| Batch Type | Token | Token |
| Batch Size | 10240 | 10240 |
| Gradient Accumulation Steps | 1 | 1 |
| Training Steps | 40000 | 200000 |
| Dropout | 0.0 | 0.1 |
| Attention Dropout | 0.1 | 0.0 |
| Max Gradient Norm | 0.5 | 0.5 |
| Warmup Steps | 8000 | 6000 |
| Decay Method | noam | noam |
| Label Smoothing | 0.1 | 0.1 |
| Layer Number | 6 | 6 |
| Head Number | 4 | 8 |
| Learning Rate | 5.0 | 5.0 |
| Adam $\beta_2$ | 0.998 | 0.998 |

