# OpenReview forum: "Riemannian SAM: Sharpness-Aware Minimization on Riemannian Manifolds"
_NeurIPS.cc/2023/Conference — NeurIPS 2023 poster_

### Official Review · Reviewer_nQre · 2023-07-03

**Soundness:** 4 excellent
**Presentation:** 3 good
**Contribution:** 3 good
**Rating:** 6
**Confidence:** 3

**Summary:**

This paper takes the popular idea of SAM from Euclidean space to the Riemannian space, and proposed a new optimization framework known as Riemannian SAM, which is a generalization of an existing technique. At the same time, the authors provided a convergence analysis of this newly proposed framework.

**Strengths:**

1. The authors proposed a general framework that is derived from first-principles, making this paper easy to follow, and provides users with sufficient background to understand.

2. Algorithm 1 is presented nicely, and makes a lot of sense under the SAM framework.

3. The convergence analysis is much needed, as it ensures readers that the new Riemannian SAM is not that much more worse than the Euclidean SAM.

4. Experiments are convincing, and are performed appropriately.

**Weaknesses:**

1. We need a bit more motivation on why riemannian optimization needs SAM, as some optimization problems can be solved both in the euclidean space and the Riemanian space (say matrix sensing), and if they have similar guarantees, users would obviously avoid constrained optimization. My understanding is that for these problems, the additional structures that Riemannian optimization brings is already important, and SAM might not help a lot. I understand the authors tried to make this point by using the two examples, but it is better to have some theoretical explanations. Also this paper would benefit a lot from explaining why (7) is the best SAM formulation for Riemannian optimization, as it just seems like a direct transport form the euclidean problem.

2. Not a lot of new insights and or proof techniques are proposed in this paper, as it seems like the proof procedures are very much alike the euclidean problem, and only adapted to the riemannian regime by assuming a chain of conditions (C-1) to (C-5). Therefore no specific new observations are made regarding the reimannian problem, and the authors are merely adapting the existing framework to "make it work" in riemannian space.

**Questions:**

N/A

---

> ### Author Rebuttal · Authors · 2023-08-10
>
> Thanks for your constructive feedbacks.
>
> **[On Motivation on Riemannian SAM formulation]**
>
> We believe that considering sharpness-aware minimization in Riemannian optimization could help the optimization. The "flat minima hypothesis" in Euclidean space was put forth, suggesting that neural networks achieve better generalization when they are trained to converge to flatter regions within the loss landscape. In this sense, the Sharpness-Aware Minimization (SAM), which encourages flat minima by (informally) minimizing the Euclidean gradient norm, has been demonstrated to effectively enhance model generalization in large-scale deep learning scenarios such as Vision Transformer and MLP-Mixer. Building on this, due to the fact that an $n$-dimensional manifold locally bears a resemblance to an $n$-dimensional Euclidean space, we expect that the flat minima hypothesis proposed in Euclidean space would hold to some extent for objective functions defined on general manifolds. The Riemannian SAM algorithm can be informally understood as reducing the Riemannian gradient norm $\lVert \mathrm{grad} f(w)\rVert_w$, which reflects the local curvature and geometry of the manifold (hence, the underlying structure of the data) at the point $w$.
>
> In addition, for some optimization problems that can be solved via both Euclidean SAM and Riemannian SAM, the Euclidean algorithm might not adequately take into account the underlying geometry of the manifold.
>
> In order to validate our intuition, we address the challenge of optimizing a neural-net-like objective function defined on the unit sphere manifold $\mathbb{S}^{2}$.
>
> The synthetic dataset is generated by drawing a total of $500$ samples from a standard Gaussian distribution $\mathcal{N}(0, 1^2)$ for $X$ and a uniform distribution $\mathcal{U}(0,1)$ for $y$, resulting in $X\in\mathbb{R}^{500 \times 3}$ and $y \in\mathbb{R}^{500}$. We choose a non-linear regression MSE loss, specifically $f(w)=\frac{1}{2n}\lVert y - \mathrm{ReLU}(Xw)\rVert_2^2$. In order to craft an objective function on the unit sphere, we impose the constraint $\mathcal{C} = \lbrace w\in\mathbb{R}^3: \lVert w\rVert_2=1\rbrace$ on the model parameter $w\in \mathbb{R}^3$.
>
> The Figure 1-(a) in the attached PDF file corresponds to Cartesian coordinates $w=(x,y,z)$ to spherical coordinates $(r,\theta,\varphi)=(1,\theta,\varphi)$, rendering contour plots. In Figure 1-(a) in the attached PDF file, we showcases the converged points on the objective function under the optimization using Riemannian SAM (in purple color) and the conventional Euclidean SAM (in pink color).
>
> Within a maximum iteration budget $100$, we choose hyperparameters for each optimization algorithm. In Figure 1-(a), the purple point (Riemannian SAM) attains a loss value of $0.3800$ while the pink point (conventional Euclidean SAM) converges with a slightly higher loss value of $0.3808$.
>
> Furthermore, in terms of sharpness measures, we considered the following basic two quantities: (i) the trace of the Hessian (sharpness in the context of Euclidean space), and (ii) Manifold-Aware Sharpness, characterized by the Riemannian gradient norm $\lVert\mathrm{grad} f(w)\rVert_w$. Notably, Manifold-Aware Sharpness aligns with Information-Geometric Sharpness [JLP+22] when dealing with statistical manifolds, where the Riemannian metric is defined by the Fisher information. For the aforementioned problem, we compare two metrics and the Figure 1-(b,c) in the attached PDF file depict the results. As seen in Figure 1-(b,c), Riemannian SAM achieves smaller sharpness values than Euclidean SAM, implying convergence toward flatter regions.
>
> In other words, since the Euclidean SAM might fail to properly consider the underlying structure of the manifold even for toy examples, this phenomenon is expected to be exacerbated in extremely high-dimensional problems such as deep learning.
>
> **Reference**
>
> - [JLP+22] A Reparametrization-Invariant Sharpness Measure Based on Information Geometry, NeurIPS 2022.
>
> ---
> **[On Theoretical Insights]**
>
> Due to the space constraint of rebuttals, we answer this concern for theory in general response. Please refer to “**on theoretical side**” in **1. Riemannian SAM is a non-trivial extension of Euclidean SAM with novel theoretical insights** in general response.

---

> > ### Comment · Reviewer_nQre · 2023-08-16
> >
> > Thank you for your detailed response, and I appreciate your time and effort for preparing these results.
> >
> > I understand that there may be benefits to using SAM in the Riemannian setting, but your paper including your rebuttal did not give me any high-level intuition on why it is important, or how it affects the optimization landscape in a different way comparing to the Euclidean setting (or is identical, that is). I am saying that this paper still lacks insights that will benefit a wider range of audience, which is the most important part of conference papers.
> >
> > However, I do agree that this framework has a lot of its own benefits, and the authors did a great job in presentations and derivations, and I do believe NeuRIPS would benefit from having this paper, therefore I am keeping my original score.

---

> > > ### Author Response · Authors · 2023-08-19
> > > **Response to the reviewer**
> > >
> > > Thanks for your valuable comments.
> > >
> > > As reviewer's suggestion, in the revision, we will clarify our contributions of our paper incorporating the intuition on the Riemannian SAM in terms of both theory and pratice.

---

### Official Review · Reviewer_wvpH · 2023-07-07

**Soundness:** 3 good
**Presentation:** 3 good
**Contribution:** 3 good
**Rating:** 6
**Confidence:** 4

**Summary:**

The authors propose an extension of the sharpness-aware minimization (SAM) technique on Riemannian manifolds, for which specific computations are feasible as the retraction map and the vector transport. The proposed optimization method considers the nonlinear geometry that is implied due to the manifold when performing a step. The convergence of the method is theoretically proven under assumptions, while empirical results are included to demonstrate the efficiency of the method.

**Strengths:**

- The extension of SAM to Riemannian manifolds is natural and meaningful.
- Theoretical guarantees for convergence are provided.

**Weaknesses:**

- Even if the paper is in general well-written, there are parts where the presentation is quite high-level and thus becomes unclear. (See questions for specific examples)
- Perhaps some figures could have been used to make the paper more accessible and easy to understand.
- I think that the experimental section implicitly assumes familiarity with the associated techniques, which makes the paper not-self-contained. Perhaps, some details about the settings could have been included in the appendix e.g. the actual models and the corresponding parameter spaces, the retraction maps, the vector transports, etc.
- As the performance of the proposed method seems to be close to the Riemannian Adam technique, I believe that at least some error-bars could have been included to better justify the difference in performance.

**Questions:**

Q1. It is not clear in many places if the manifold $\mathcal{M}$ is considered as being embedded within an ambient (Euclidean) space or as (a subset of) $\mathbb{R}^D$ together with a Riemannian metric as in Fisher SAM. In the embedded case, typically the Euclidean gradient is computed, which is then projected orthogonally to the tangent space followed by a retraction map. In the other case, the gradient is the Euclidean gradient multiplied by the inverse of the metric, while for the update step, simple addition is used.

Q2. Related to Q1. The steps in lines 146-147 are quite confusing. Is it implied here that the actual manifold on which $\mathcal{L}$ is defined is actually a surface embedded within an ambient Riemannian manifold where the metric $g(w)$ defines the structure? If this is the case, then the "orthogonal" projection on the tangent space should be defined with respect to the ambient Riemannian metric?

Q3. Related to Q1 & Q2. In Eq 15, could you explain based on Q1 and Q2 why we need to use both the inverse of the metric and the projection to compute the gradient?

Q4. Is the Stiefel manifold used somewhere as it is presented in line 130?

Q5. In Eq. 12 it is mentioned that higher-order Riemannian gradient is necessary. Could you elaborate on which computation is infeasible? Due to the chain-rule the derivative of the retraction map with respect to the base point $w$?

Minor:
- Line 202-203: Something is missing at the end of the sentence.
- Line 210: Probably $\eta \in \mathcal{T}_w\mathcal{M}$ .
- Eq 15: probably the $\nabla$ is missing.
- Line 164: Capital S at the begging of the sentence.
- Line 122: Is the function $\phi$ an isometry?
- Line 101: Only isometric vector transport implies that the angle between tangent vectors may change? Any influence on the optimization?

**Limitations:**

The authors discuss some limitations of their work, such as the computational cost, while the standard limitation that comes with such geometric approaches is the access to the associated manifold operations e.g. retraction, vector transport, etc. As a theoretical work, it does not have any direct negative societal impact.

---

> ### Author Rebuttal · Authors · 2023-08-10
>
> Thanks for your constructive feedbacks.
>
> **[On Q1 and Q2]**
>
> As the reviewer pointed out, in most practical cases, the manifolds under consideration are often embedded in Euclidean space or are subsets of $\mathbb{R}^d$ with an appropriate Riemannian metric. For instance, in the case of one of an elementary manifold, the $n$-dimensional hypersphere $\mathbb{S}^{n-1}$, it is embedded submanifold of Euclidean space. Thus, the inner product defined on the tangent space $T_x \mathbb{S}^{n-1}$ for any point on the hypersphere $x\in \mathbb{S}^{n-1}$ is simply the Euclidean inner product (i.e., $\langle u, v\rangle_x = u^\mathsf{T} v$ for any $x$ and any tangent vectors $u,v\in T_x \mathbb{S}^{n-1}$). As the reviewer mentioned, the computations of gradients on the hypersphere involve calculating the Euclidean gradient first and then projecting it onto the tangent space. Furthermore, in deep learning on manifolds, prevalent manifold structures include the Poincaré ball and Lorentz model on the hyperbolic space, along with the Stiefel manifold that encourages orthogonality among parameters. All three of these manifolds are defined on subsets of $\mathbb{R}^d$ with suitable Riemannian metrics. In this case, the computation of the Riemannian gradient, as the reviewer described, involves preconditioning the Euclidean gradient with the inverse metric.
>
> ---
> **[On Q3]**
>
> Indeed, the Lorentz model is not a Riemannian manifold but rather a (semi)-Riemannian manifold. Nevertheless, Lorentz model remains amenable to Riemannian optimization. In consequence, due to the absence of a guarantee that the Euclidean gradient preconditioned with the inverse metric (i.e., the Riemannian gradient) resides on the tangent space, the projection operation onto the tangent space becomes necessary. The equation (8) in our paper illustrates this projection procedure.
>
> ---
> **[On Q4]**
>
> While the Stiefel manifold was not considered in our experiments in this paper, it is mainly employed to encourage ***parameter orthogonality***. The several studies have shown that parameter orthogonality influences model generalization and serves as a remedy for the vanishing/exploding gradient problem. We include some references below.
>
> **Reference**
>
> - [LJW+21] Orthogonal Deep Neural Networks, TPAMI 2021.
>
> - [TK21] Orthogonalizing Convolutional Layers with the Cayley Transform, ICLR 2021 Spotlight.
>
> - [FT21] Efficient Riemannian Optimization on the Stiefel Manifold via the Cayley Transform, ICLR 2020.
>
> - [WCC20] Orthogonal Convoluiontal Neural Networks, CVPR 2020.
>
> ---
> **[On Q5]**
>
> Yes, you are right. The retraction already involves the model parameter $w$, hence we should apply chain rule with respect to $w$, which requires the (infeasible) higher-order Riemannian gradient in practice.
>
> ---
> **[On Minor Points]**
>
> Thanks for the correction of typos in the paper. In line 122, $\varphi$ is known to be isometric. In another context, the isometricity of the vector transport exhibits the characteristics of symmetry/angle preservation and also the preservation of vector lengths. These attributes facilitate the theoretical analysis of Riemannian optimization and the isometricity of vector transports stands as one of the standard assumptions in Riemannian optimization analysis.

---

> > ### Comment · Reviewer_wvpH · 2023-08-21
> > **Post-rebuttal**
> >
> > I would like to thank the authors for their replies. After considering the other reviews, I tend to agree that the current paper is simply the Riemannian version of SAM, apart from the technical differences. However, I think that this is a fair contribution and for this reason, I will keep my score. I suggest the authors taking into account the reviews and update the paper accordingly.

---

### Official Review · Reviewer_zxyx · 2023-07-10

**Soundness:** 3 good
**Presentation:** 3 good
**Contribution:** 2 fair
**Rating:** 3
**Confidence:** 4

**Summary:**

This paper introduce the new objective function SAM (Shapeness-awared minimization) on optimization problems on Riemannian manifolds. The motivation is that, with the success of SAM on the Euclidean space, the new Riemmanian metric, which is typically different from the flat Euclidean metric, can introduce more domain prior knowledge to SAM. Example applications include social network analysis and knowledge graph completion. The paper derived a new objective function based on retraction operations on manifolds, and provide convergence analysis of the Riemannian SGD algorithm. Empirical results show slight improvement over baseselines that are not based on the SAM objective function.

**Strengths:**

The overall motivations make sense and the paper has made the contribution of combining SAM and optimization on Riemannian manifolds.

**Weaknesses:**

The main weaknesses are as follow:
1) The derivations of the new objective relies on standard tools from optimization on manifold and there is limited technical contribution.
2) Though SAM was published and proved useful, it is still not clear why minimizing a normalized norm of Riemannian gradient helps improve the test accuracy (line 190). The statement on lines 59-60 is not well supported. It is understandable that the main focus of this submission is not on renovating SAM, but some informal explanation can be helpful.
3) Only the best test accuracies are reported. The authors are encouraged to conduct cross-valiation and/or report average performance based on randomized experiments.
4) In my opinion, the improvement of the test accuracies is visible but not quite significant.

**Questions:**

line 101: should <\xi,\eta> be evaluated on the tangent space at z rather than w?
line 112: the angle is not formally defined.

---

> ### Author Rebuttal · Authors · 2023-08-10
>
> Thanks for your constructive feedbacks.
>
> **[On Technical Limitations]**
>
> We appreciate your valuable feedbacks, but we do not agree with the reviewer’s opinion. Due to space constraint of rebuttals, we answer this concern in **1. Riemannian SAM is a non-trivial extension of Euclidean SAM with novel theoretical insights** in general response.
>
> ---
> **[On Intuition of Riemannian SAM]**
>
> The "flat minima hypothesis" in Euclidean space was put forth, suggesting that neural networks achieve better generalization when they are trained to converge to flatter regions within the loss landscape. In this sense, the Sharpness-Aware Minimization (SAM), which encourages flat minima by (informally) minimizing the Euclidean gradient norm, has been demonstrated to effectively enhance model generalization in large-scale deep learning scenarios such as Vision Transformer and MLP-Mixer.
>
> Building on this, due to the fact that an $n$-dimensional manifold locally bears a resemblance to an $n$-dimensional Euclidean space, we expect that the flat minima hypothesis proposed in Euclidean space would hold to some extent for objective functions defined on general manifolds. The Riemannian SAM algorithm can be informally understood as reducing the Riemannian gradient norm $\lVert \mathrm{grad} f(w)\rVert_w$, which reflects the local curvature and geometry of the manifold (hence, the underlying structure of the data) at the point $w$.
>
> In order to validate our intuition, we address the challenge of optimizing a neural-net-like objective function defined on the unit sphere manifold $\mathbb{S}^{2}$.
>
> The synthetic dataset is generated by drawing a total of $500$ samples from a standard Gaussian distribution $\mathcal{N}(0, 1^2)$ for $X$ and and a uniform distribution $\mathcal{U}(0,1)$ for $y$ respectively, resulting in $X\in\mathbb{R}^{500 \times 3}$ and $y\in\mathbb{R}^{500}$. We choose a non-linear regression MSE loss, specifically $f(w)=\frac{1}{2n}\Vert y - \mathrm{ReLU}(Xw) \Vert_2^2$. In order to craft an objective function on the unit sphere, we impose the constraint $\mathcal{C} = \lbrace w\in\mathbb{R}^3: \lVert w \rVert_2=1 \rbrace$ on the model parameter $w \in \mathbb{R}^3$.
>
> The Figure 1-(a) in the attached pdf file corresponds to Cartesian coordinates $w=(x,y,z)$ to spherical coordinates $(r,\theta,\varphi)=(1,\theta,\varphi)$, rendering contour plots. In Figure 1-(a), we showcases the converged points on the objective function under the optimization using Riemannian SAM (in purple color) and the conventional Euclidean SAM (in pink color).
>
> Within a maximum iteration budget $100$, we search best hyperparameters for each optimization algorithm. In Figure 1-(a), the purple point (Riemannian SAM) attains a loss value of $0.3800$ while the pink point (conventional Euclidean SAM) converges with a slightly higher loss value of $0.3808$.
>
> Furthermore, in terms of sharpness measures, we considered the following basic two quantities: (i) the trace of the Hessian (sharpness in the context of Euclidean space), and (ii) Manifold-Aware Sharpness, characterized by the Riemannian gradient norm $\lVert \mathrm{grad} f(w)\rVert_w$. Notably, Manifold-Aware Sharpness aligns with Information-Geometric Sharpness [JLP+22] when dealing with statistical manifolds, where the Riemannian metric is defined by the Fisher information. For the aforementioned problem, we compare two metrics and Figure 1-(b,c) depict the results. In both metrics, Riemannian SAM achieves smaller sharpness values than Euclidean SAM, implying convergence toward flatter regions.
>
> In other words, since the Euclidean SAM might fail to properly consider the underlying structure of the manifold even for toy examples, this phenomenon is expected to be exacerbated in extremely high-dimensional problems such as deep learning.
>
> **Reference**
>
> - [JLP+22] A Reparametrization-Invariant Sharpness Measure Based on Information Geometry, NeurIPS 2022.
>
> ---
> **[On Experiments]**
>
> Contrary to the reviewer’s belief, we believe that our improvement is significant compared to baselines. For detailed response, due to space constraint of rebuttals, we answer the concerns in “**2. On Experiments”** in general response and include the tables for the additional experiments in the attached PDF file.

---

### Official Review · Reviewer_SuW3 · 2023-07-17

**Soundness:** 3 good
**Presentation:** 2 fair
**Contribution:** 2 fair
**Rating:** 4
**Confidence:** 4

**Summary:**

 This paper extends the Sharpness-Aware Minimization (SAM) algorithm to Riemannian manifolds. They show that the new model subsumes Fisher SAM as a special case, and it leads to a new algorithm (called Lorentz SAM when specified to the Lorentz manifold). The main proposed algorithm is given in Algorithm 1, which works for general Riemannian manifolds, and a convergence theorem is provided for it (Theorem 1). The paper is concluded with some experiments showing the gain of the new algorithm in various settings.


**Strengths:**


- novel formulation allowing to extend SAM to Riemannian manifolds (Algorithm 1)

- theoretically verified model (Theorem 1)

- various experiments




**Weaknesses:**

- contributions of this paper are unfortunately limited to a few examples

- the proposed algorithm mimics the original SAM after making necessary changes

- Algorithm 1 is not well explained, though it's the main contribution of the paper





**Questions:**



An interesting result on optimization on manifolds; but I'm concerned about the sufficiency of the contributions of the paper.  Besides that, the method is not well explained and the paper could've been better written.




Questions/Comments:

- Line 117: "Lorentz model is a Riemannian manifold" This is wrong since the Lorentz manifold is a pseudo-Riemannian manifold, as the metric is not positive-definite. I'm wondering how this can change the algorithm's proof and applicability to the Lorentz manifold.




- Line 145-147: This is only true when there is a nice extension of the function on the Euclidean space, and so might not be always feasible.

 - Algorithm 1: the role of the base optimizer is not well explained.

 - Line 8: what is the role of the transportation? Why can't one update the parameters after the ascent step (then one does not need transportation?)? This part is also not well explained

---

> ### Author Rebuttal · Authors · 2023-08-10
>
> Thanks for your constructive feedbacks.
>
> **[On Contributions]**
>
> Thank you for your important feedbacks, but we believe that our main example, hyperbolic representation learning, is not a simple “one example” but has a sufficiently general use cases, and that proposing an optimization technique that can advance it can be a significant contribution to the community.
>
> Deep learning in conventional Euclidean space often falls short in effectively capturing hierarchical relationships, efficiently representing tree-like structures, addressing dimensionality challenges (ex. embeddings), and etc. Under such limitations, one of main approach in geometric deep learning, ***hyperbolic representation learning***, has gained significant importance in various fields due to its ability to capture such relations in various data, for example, images, languages, graphs, and so on. The hyperbolic representation learning has been able to address important challenges in Euclidean space more intelligently by taking into account the underlying geometric characteristics of the data, thereby enabling us to devise significantly advanced techniques. We introduce several previous studies.
>
> While it holds true that the majority of modern deep learning architectures and techniques continue to be proposed in Euclidean space, the extension of successful models and foundational approaches from Euclidean to (Riemannian) manifolds, irrespective of the domain, is gaining attention. Moreover, (Riemannian) manifold extensions that consider the underlying data structure often outperform their Euclidean counterparts. For instance, in computer vision fields, the methodologies extended to hyperbolic spaces [VLB+19, KMU+20, GMK23, SKW+23, SBM23] have been actively studied for the important problem including image embedding, classification, segmentation, and etc.
>
> Also, the well-regarded generative models in Euclidean space such as normalizing flow and diffusion models have successfully been extended to Riemannian Manifolds [LFC21, MN20, HAB+22].  Notably, even in language modeling, Transformer which is a main backbone, has been extended with fully hyperbolic modules [CHL+22], and hyperbolic Transformer outperforms the Euclidean counterpart. Moreover, in this year, the pioneering research [CCB+23] that incorporated hyperbolic geometry into deep reinforcement learning has been published in ICLR as a spotlight. More recently, avenues for enhancing numerical stability in hyperbolic representation learning have been published [MWW+23] in ICML 2023.
>
> In this sense, we do not perceive our contribution to be limited solely to a few examples. Rather, we believe that our optimization approach has a great potential for broader applicability across various domains on manifold-aware deep learning.
>
> **References**
>
> - [VLB+19] Manifold Mixup: Better representations by Interpolating Hidden States, ICML 2019
>
> - [KMU+20] Hyperbolic Image Embeddings, CVPR 2020
>
> - [ELM+19] Continuous hierarchical representations with poincare ́ variational auto-encoders, NeurIPS 2019
>
> - [LFC21] Hyperbolic Generative Adversarial Networks, IEEE Access 2021
>
> - [MN20] Riemannian Continuous Normalizing Flows, NeurIPS 2020
>
> - [HAB+22] Riemannian Diffusion Models, NeurIPS 2022
>
> - [GSA+22] Hyperbolic Image Segmentation, CVPR 2022
>
> - [CHL+22] Fully Hyperbolic Neural Networks, ACL 2022
>
> - [GMK23] Hyperbolic Contrastive Learning for Visual Representations beyond Objects, CVPR 2023
>
> - [SKW+23] Robust Hierarchical Symbolic Explanations in Hyperbolic Space for Image Classification, CVPR 2023
>
> - [CCB+23] Hyperbolic Deep Reinforcement Learning, ICLR 2023
>
> - [MWW+23] The Numerical Stability of Hyperbolic Representation Learning, ICML 2023
>
> - [SBM23] Poincare ResNet, ArXiv 2023
>
> ---
> **[On Riemannian SAM formulation]**
>
> Due to space constraint of rebuttal, we answer the reviewer’s concern in **1. Riemannian SAM is a non-trivial extension of Euclidean SAM with novel theoretical insights** in general response.
>
> ---
> **[On Algorithm 1]**
>
> In Algorithm 1, the term “base optimizer $\mathcal{A}$” refers to any optimizer that can be employed on a manifold, such as Riemannian SGD or Riemannian Adam. Regarding line 9 in Algorithm 1, $\Delta_t^{adv}$ signifies the final update vector constructed using A with Riemannian SAM gradient. In other words, the Riemannian SAM gradient $g_t^{adv}$ (line 8 in Algorithm 1) can be utilized to construct the first-order/second-order momentum of Riemannian Adam. Or alternatively, it can be directly employed for performing Riemannian SGD. We will clarify it.
>
> ---
> **[On Lorentz Model]**
>
> This is a mistake in our statement, and we apologize for the confusion. As pointed out by the reviewer, while it is true that the Lorentz model involves a (semi)-Riemannian manifold, we can still employ the Riemannian optimization introduced in our paper to optimize the objective function defined on this manifold. In light of this context, we will provide a clearer description in the revision.
>
> ---
> **[On Role of Vector Transport in Line 8]**
>
> In Algorithm 1 and according to our Riemannian SAM formulation, the Riemannian gradient computed at $w=w_t^{adv}$ is not a vector on the tangent space $T_{w_t} \mathcal{M}$ defined at the point $w=w_t$, making the algebraic operations impossible for directly updating $w_t$. Consequently, in order to update the original parameter $w_t$ using the Riemannian gradient computed at the perturbed point, an operation is required to bring it into the tangent space at $w_t$, which can be achieved via the vector transport $\mathcal{T}_{w_t^{adv}}^{w_t}$  at line 8 in Algorithm 1.

---

> > ### Comment · Reviewer_SuW3 · 2023-08-14
> > **Response**
> >
> > I acknowledge the response provided by the authors. I still suggest revising the paper to clarify the contributions. Please also apply my comment on the Lorentz model to the revised version. Given the authors' efforts and after reading the responses and the general comment, I decided to slightly increase my score (from 3 to 4). Thanks!

---

> > > ### Author Response · Authors · 2023-08-14
> > > **Thank you for your comments and increasing the score. We are wondering if there is any further suggestions.**
> > >
> > > We sincerely appreciate your comments as they have strengthened our paper and also thank the reviewer for increasing the score.
> > > In the revised version, we assure that we will further clarify the contributions of our paper and incorporate your comments regarding the Lorentz model.
> > >
> > > As the reviewer mentioned that our paper still needs to be revised, we are wondering if there might be any further suggestions you could provide on potential avenues for improving our paper.

---

### Author Rebuttal · Authors · 2023-08-10

Due to the space constraints on each rebuttal, we answer some important questions in general response here.

**1. Riemannian SAM is a non-trivial extension of Euclidean SAM with novel theoretical insights**

* **[On technical side]**

We would like to emphasize that our Riemannian SAM does NOT merely mimic the original SAM either theoretically or methodologically. In regarding methodology, there could be several extensions of conventional Euclidean SAM to a Riemannian optimization in different manners.

For example, it is most natural to choose a perturbation region at the current point $w_t$ as in the conventional Euclidean SAM, $\delta \in B_\rho(w_t) = \lbrace x \in \mathcal{M}: d_\mathcal{M}(w_t,x) \leq \rho \rbrace$ where $d_\mathcal{M}$ represents the distance on the manifold. However, adopting the constraint on $\delta$ in this manner may pose challenges in utilizing the standard assumptions for analyzing non-convex Riemannian optimization, such as geodesic or retraction smoothness (see condition (C-4) in our paper), which makes it difficult to guarantee convergence. Moreover, the computation of $d_\mathcal{M}$ is often computationally inefficient in practice. Another possible extension is to apply the vector transport operation from Equation 8 of Algorithm 1 to Equation 9. The following outlines the modified procedure. (i) $g_t^{adv} = \mathcal{A}(grad \mathcal{L}(w; \mathcal{S})\lvert_{w=w_t^{adv}})$ and (ii) $\Delta_t^{adv} = \mathcal{T}_{w_t^{adv}}^{w_t} g_t^{adv}$.

For base optimizer $\mathcal{A}$, any optimization algorithm commonly used in Riemannian optimization can be adopted (e.g., Riemannian SGD, Riemannian momentum, etc.). However, when vector transport is applied after constructing $g_t^{adv}$ via the momentum-based optimizer $\mathcal{A}$, the momentum construction takes place on the tangent space $T_{w_t^{adv}} \mathcal{M}$ at the perturbed point $w_t^{adv}$, while the parameter update occurs on the different tangent space $T_{w_t} \mathcal{M}$ at the point $w_t$. As a result, this might introduce another challenges in understanding and analyzing the overall optimization process.
    - In this perspective, various alternative extensions are also possible, but among them, we have carefully designed a ***theoretically valid, computationally practical, and non-trivially extended Sharpness-Aware Minimization to a manifold for Riemannian optimization.*** Then, we have successfully demonstrated both convergence analysis and empirical studies to corroborate our Riemannian SAM.

* **[On theoretical side]**

In terms of theory, our key observation of Theorem 1 in the paper lies in ***the alignment between the Riemannian SAM gradient $\mathcal{T}_{w_t^{adv}}^{w_t} \mathrm{grad} f(w_t^{adv})$ and the Riemannian gradient $\mathrm{grad} f(w_t)$*** for the perturbation point, $w_t^{adv} = R_{w_t}(\rho_t \mathrm{grad}f(w_t))$. The previous study [50] on Euclidean SAM says that Euclidean SAM gradient should be well-aligned with the true gradient step for convergence. Unlike the theoretical claim in [50], we stress that for convergence guarantee those gradients should be well-aligned within the preconditioned space (by inverse Riemannian metric) regardless of alignment in Euclidean space.

To verify this insight, we directly measure the angles between two vectors with a 2D toy example, illustrating how they align in practice. Toward this, we consider two angles: (i) $\angle(\nabla f(w_t^{adv}), \nabla f(w_t))$ (Euclidean Alignment) and (ii) $\angle (\mathcal{T}_{w_t^{adv}}^{w_t} \mathrm{grad} f(w_t^{adv}), \mathrm{grad} f(w_t))$ (Riemannian Alignment, Ours)

In this example, we consider the logistic regression where $200$ data samples are generated with $100$ of them sampled from $\mathcal{N}(-1, 1^2)$ and the remaining $100$ sampled from $\mathcal{N}(1, 1^2)$. The labels are assigned such that if a sample was drawn from a Gaussian distribution with a mean of $-1$, the label was set to $y=0$, and otherwise, we set $y=1$. We minimize the cross-entropy loss with our Riemannian SAM with the Fisher information matrix as the Riemannian metric.

The Figure 2 in the attached PDF file depicts the comparison of angles. The loss decreases up to 10-th iteration, after which it remains around the converged point. As evident from the illustration, while the angles between the Euclidean space SAM gradient and the gradient deviate by up to around 25 degrees, the angles between the preconditioned SAM gradient and the preconditioned gradient, influenced by the Fisher information, align more closely with deviations only up to a maximum of 10 degrees. In high-dimensions, we expect that the angles would become significantly larger, corroborating our theoretical insight.

**2. On Experiments**

We did not select the best model based on the test data; instead, we opt for the best model based on the validation loss. Some baselines have reported their best test performance, which might have led reviewers to misunderstand our results. In fact, we report the average performance for knowledge graph completion and choose hyperparameters based on validation. Thus, our evaluation is comparatively more stringent when contrasted with those previous studies, placing us at a potential disadvantage in terms of evaluation. Also, as the reviewer's suggestion, we conduct additional experiments by varying seeds. Please refer to Table 1 in the attached PDF file.

During the rebuttal period, we conduct additional experiments with expanded size of Transformer on machine translation tasks. Given this large-scale experiments, it was hard to run randomized simulations for the baselines as well, due to rebuttal time constraints. As a result, we report test BLEU scores by selecting the model *based on validation loss*. Please refer to Table 2 in the attached PDF file.

According to Table 1 and 2 in the attached PDF file, we emphasize that our results are NOT marginal for all experiments considered in our paper.

---

### Decision · Program_Chairs · 2023-09-21

**Decision:**

Accept (poster)

**Comment:**

This paper proposes an extension of the sharpness-aware minimization (SAM) technique on Riemannian manifolds, for which specific computations are feasible as the retraction map and the vector transport. The proposed optimization method considers the nonlinear geometry that is implied due to the manifold when performing a step. This extension is non-trivial because there are several other design choices, but for the extension presented in the current paper, the authors are able to prove convergence under related assumptions, which is the main strength of the paper.  This paper also demonstrated the efficiency of the method via empirical results.

One main weakness is the inadequate discussion on the intuition of performing SAM on riemannian manifold, as pointed out by the reviewers. I suggest that the authors carefully revise the paper according to the various useful suggestions given by the reviewers.